# Visualizing PIEZO1 localization and activity in hiPSC-derived single cells and organoids with HaloTag technology

Gabriella A. Bertaccini [1,2], Ignasi Casanellas [1,2], Elizabeth L. Evans[1,2], Jamison L. Nourse[1,2], George D. Dickinson [1], Gaoxiang Liu[3], Sayan Seal[3], Alan T. Ly [1,2], Jesse R. Holt [1,2,4], Tharaka D. Wijerathne [5], Shijun Yan [6], Elliot E. Hui [6], Jerome J. Lacroix[5], Mitradas M. Panicker[1,2], Srigokul Upadhyayula [3,7,8], Ian Parker[1,9] & Medha M. Pathak [1,2,4,6] ✉

PIEZO1 is critical to numerous physiological processes, transducing diverse mechanical stimuli into electrical and chemical signals. Recent studies underscore the importance of visualizing endogenous PIEZO1 activity and localization to understand its functional roles. To enable physiologically and clinically relevant studies on human PIEZO1, we genetically engineered human induced pluripotent stem cells (hiPSCs) to express a HaloTag fused to endogenous PIEZO1. Combined with advanced imaging, our chemogenetic platform allows precise visualization of PIEZO1 localization dynamics in various cell types. Furthermore, the PIEZO1-HaloTag hiPSC technology facilitates the non-invasive monitoring of channel activity across diverse cell types using $Ca^{2+}$-sensitive HaloTag ligands, achieving temporal resolution approaching that of patch clamp electrophysiology. Finally, we use lightsheet microscopy on hiPSC-derived neural organoids to achieve molecular scale imaging of PIEZO1 in three-dimensional tissue. Our advances establish a platform for studying PIEZO1 mechanotransduction in human systems, with potential for elucidating disease mechanisms and targeted drug screening.

PIEZO channels are pivotal in transducing mechanical stimuli into electrical and chemical signals, and play a significant role in a wide range of physiological functions[1,2]. PIEZO1, in particular, is expressed across various excitable and non-excitable tissues, shaping key biological processes such as vascular development[3,4], exercise physiology[5], blood pressure regulation[6–8], red blood cell volume regulation[9,10], neural stem cell differentiation[11,12], and wound healing[13,14]. The channel has been linked to several human diseases, including hereditary xerocytosis[15–18], lymphatic dysplasia[19,20], iron overload[21], and malaria[22],

and ongoing studies on this recently-identified protein are likely to uncover more disease associations.

So far, the study of PIEZO1 channel function has primarily utilized patch clamp electrophysiology[1,23–25] and, more recently, measurements of $Ca^{2+}$ influx through the channel[11,26–30]. These methodologies have provided valuable insights into the biophysical properties of PIEZO1. However, modulation of PIEZO1 function is complex, and the nature of PIEZO1 activity, as well as its downstream outcomes, are highly context-dependent[31]. Thus, to fully understand how PIEZO1

[1]Department of Physiology and Biophysics, University of California, Irvine, CA, USA. [2]Sue and Bill Gross Stem Cell Research Center, University of California, Irvine, CA, USA. [3]Advanced Bioimaging Center, Department of Molecular and Cell Biology, University of California, Berkeley, CA, USA. [4]Center for Complex Biological Systems, University of California, Irvine, CA, USA. [5]Department of Biomedical Sciences, Western University of Health Sciences, Pomona, CA, USA. [6]Department of Biomedical Engineering, University of California, Irvine, CA, USA. [7]Molecular Biophysics and Integrated Bioimaging Division, Lawrence Berkeley National Laboratory, Berkeley, CA, USA. [8]Chan Zuckerberg Biohub, San Francisco, CA, USA. [9]Department of Neurobiology and Behavior, University of California, Irvine, CA, USA. ✉e-mail: medhap@uci.edu

orchestrates diverse physiological roles and how its malfunction leads to disease, it is crucial to study endogenous PIEZO1 in its native cellular environment. An emerging theme in PIEZO1's physiological roles is its increased localization and activity at specific cellular structures, including focal adhesions, nuclei, and cell-cell junctions[30,32–36], suggesting a subcellular spatial organization of PIEZO1-mediated $Ca^{2+}$ signals. Notably, the dynamic spatial positioning of PIEZO1 in migrating keratinocytes is a key determinant of their wound-healing capabilities[13], highlighting the importance of PIEZO1 spatiotemporal dynamics in governing physiological processes. Taken together, these findings prompt the need for new methodologies to precisely and non-invasively monitor the spatiotemporal organization and activity of endogenous PIEZO1.

Current methods for visualizing PIEZO1 localization predominantly utilize the fusion of fluorescent proteins such as GFP[37,38] or tdTomato[3,30,39], but these methods are hampered by dimness, photobleaching, and an inability to measure channel activity. For measuring PIEZO1 activity, patch clamp electrophysiology is the standard assay[1,23,40], but it disrupts cellular mechanics and offers limited insights into the channel's spatial localization. In contrast, PIEZO1 activity measurements using cytosolic $Ca^{2+}$-sensitive indicators in native cells[11,30] though spatially informative, lack specificity and can be confounded by $Ca^{2+}$ flux through other channels. In addition, non-human model organisms commonly used to explore the physiological roles of PIEZO1 may not fully recapitulate the channel's behavior in human physiology[41,42].

Here, we develop a platform to overcome these challenges and provide a versatile human-specific system to complement animal studies, thus advancing physiologically- and clinically-relevant research on human PIEZO1. Utilizing CRISPR engineering, we introduced a self-labeling HaloTag domain fused to endogenous PIEZO1 in human induced pluripotent stem cells (hiPSCs), which can be differentiated into a variety of specialized cells and tissue organoids. Combined with the use of bright and photostable Janelia Fluor (JF)-based HaloTag ligands (HTLs)[43–45], super-resolution imaging approaches, and automated image analysis pipelines, our platform allows the study of endogenous human PIEZO1 through imaging assays in various hiPSC-derived cell types and in vitro tissue organoids. This advance not only enables quantitative imaging of PIEZO1 channel localization and activity across diverse physiological scenarios but also facilitates human disease modeling of PIEZO1. Our work opens avenues towards mechanistic studies of endogenous human PIEZO1 in a variety of cell types, disease conditions, as well as large-format drug screening for the development of targeted therapeutics.

## Results
### Development and validation of PIEZO1-HaloTag hiPSC lines
To create a multifaceted tool for visualizing endogenous PIEZO1, we utilized CRISPR engineering to tag the endogenous channel with HaloTag, a modified bacterial haloalkane dehalogenase that covalently binds to exogenously provided chloroalkane HaloTag ligands[46]. We performed the genetic edits at both copies of *PIEZO1* in hiPSCs, which are capable of self-renewal as well as differentiation into a variety of specialized cell types and tissue organoids (Fig. 1A). We attached the HaloTag to the C-terminus of PIEZO1 (Fig. 1B), as this location has previously been used for endogenous PIEZO1 tagging without affecting channel function[3]. The PIEZO1-HaloTag hiPSCs, as well as their differentiated progeny, are expected to express the HaloTag protein fused to endogenous PIEZO1, enabling covalent labeling with cognate HaloTag ligands in a variety of cell types.

We first confirmed the effective generation of the PIEZO1-HaloTag fusion protein by western blot (Supplementary Fig. 1). Using an anti-PIEZO1 antibody, we observed a band in whole-cell lysate from the parent WTC-11 hiPSCs at the expected size of approximately 289 kDa for PIEZO1 and from PIEZO1-HaloTag hiPSCs at approximately 319 kDa.

This increase in mass is consistent with the fusion of HaloTag (33 kDa) to PIEZO1. As an orthogonal assay, we performed western blotting using an anti-HaloTag antibody and observed a signal in the PIEZO1-HaloTag hiPSC line, also at approximately 319 kDa, matching the mass observed with the anti-PIEZO1 antibody. This 319 kDa protein band was absent in the parent WTC-11 line. To further confirm HaloTag incorporation to PIEZO1, we knocked out *PIEZO1* in the engineered PIEZO1-HaloTag hiPSCs. The PIEZO1-HaloTag Knockout (KO) hiPSCs did not exhibit a band using either the anti-PIEZO1 or the anti-HaloTag antibody. Taken together, we conclude that the heavier band corresponding to 319 kDa represents the endogenous PIEZO1 channel fused to the HaloTag domain and that the genetic modification results in the tagging of all PIEZO1 protein produced by the cell.

To determine whether the fusion of the HaloTag to PIEZO1 affected the channel's function, we evaluated PIEZO1 ionic currents by cell-attached patch clamp electrophysiology using hiPSC-derived endothelial cells (ECs), which show high expression of PIEZO1[3,4]. Cell-attached patch clamp measurements, with mechanical stimulation of the membrane patch through negative pressure pulses, revealed mechanically-evoked currents with slow inactivation and deactivation, as previously demonstrated in primary ECs[47]. These currents were absent in ECs differentiated from PIEZO1-Knockout WTC-11 hiPSCs, confirming that they represent ionic current through PIEZO1 (Fig. 1C). Notably, maximal currents from ECs differentiated from the PIEZO1-HaloTag hiPSCs were similar in magnitude and indistinguishable to those from ECs generated from the parent hiPSC line, indicating that the HaloTag fusion did not abrogate channel activation (Fig. 1D).

To test labeling specificity, we incubated hiPSCs with the Janelia Fluor 646 HaloTag ligand[45] (JF646 HTL, see Methods) and imaged the cells with Total Internal Reflection Fluorescence (TIRF) microscopy (Fig. 1E). The PIEZO1-HaloTag hiPSCs displayed a punctate fluorescent signal, as previously observed for endogenous, tagged PIEZO1-tdTomato channels[30]. In contrast, PIEZO1-HaloTag Knockout hiPSCs and wild-type WTC-11 hiPSCs both showed little or no punctate staining. To confirm the specific labeling of PIEZO1-HaloTag channels in the differentiated progeny of PIEZO1-HaloTag hiPSCs, we differentiated PIEZO1-HaloTag hiPSCs into three cell types which are known to express PIEZO1: ECs[3,4,39], keratinocytes[13], and neural stem cells (NSCs)[11] (Fig. 1F and Supplementary Fig. 2). Upon labeling with the JF646 HTL, each differentiated cell type exhibited a punctate fluorescent signal (Fig. 1F and Supplementary Movies 1, 2, and 3), whereas cells differentiated from PIEZO1-HaloTag Knockout hiPSCs lacked these signals (Supplementary Fig. 3). To determine whether the attachment of the HaloTag ligand affected channel activity, we performed whole-cell patch clamp on PIEZO1-HaloTag ECs and NSCs with mechanical stimulus imparted with poking using a blunt glass probe (poking assay). We compared unlabeled and HaloTag ligand-labeled samples from each cell type, and PIEZO1-HaloTag KO cells were again used as controls. The labeled cells showed no changes in inactivation or deactivation kinetics, or in maximal current amplitude, compared to unlabeled PIEZO1-HaloTag cells (Supplementary Fig. 4), indicating that labeling the PIEZO1-HaloTag channel with an HTL does not alter its functional behavior. Overall, we validated the PIEZO1-HaloTag hiPSC line as a platform to attach HaloTag ligands specifically to human PIEZO1 in multiple cell types while preserving channel function.

### PIEZO1-HaloTag imaging with high signal-to-noise and reduced photobleaching
PIEZO1 has been shown to be mobile in the plasma membrane[30,32,38,39], and we confirmed this in hiPSC-derived NSCs, keratinocytes, and ECs differentiated from PIEZO1-HaloTag hiPSCs and labeled with JF646 HTL (Supplementary Movies 1, 2, and 3). Previous studies on the mobility of endogenous PIEZO1 have utilized cells harvested from a PIEZO1-tdTomato knock-in reporter mouse[30,39], where the tdTomato fluorescence is limited by rapid photobleaching and low signal-to-

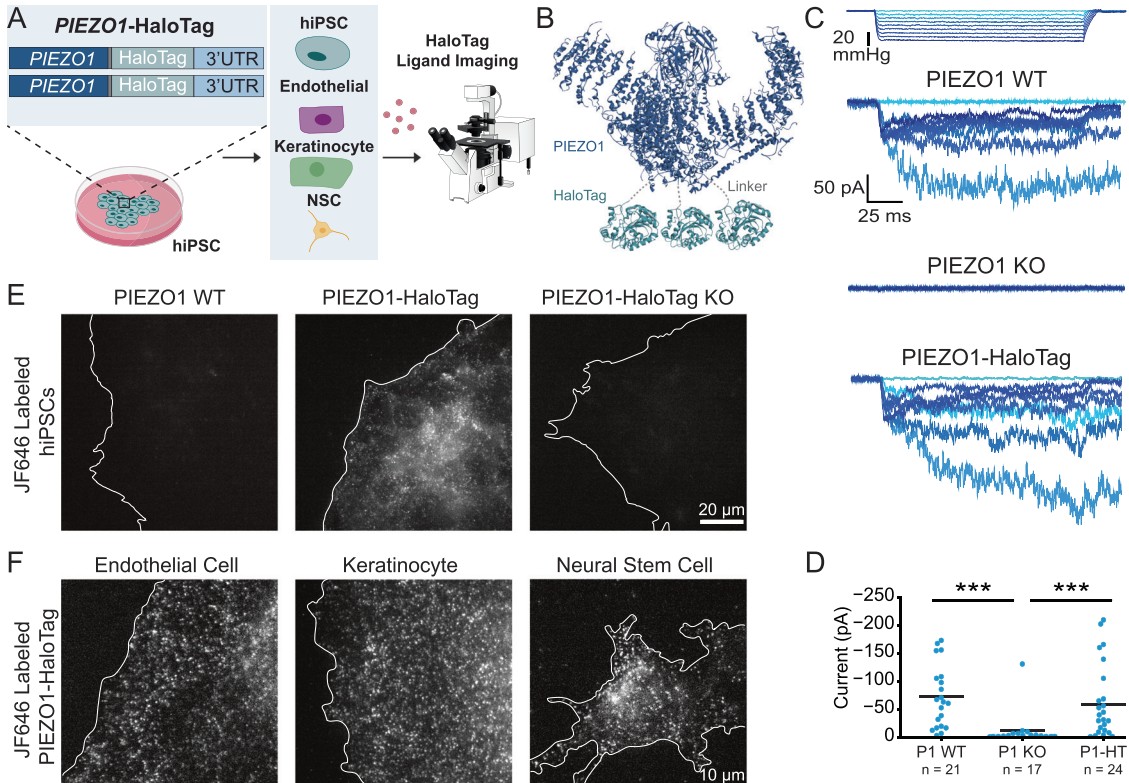

**Fig. 1 | Generation and validation of the PIEZO1-HaloTag hiPSC line. A** Flowchart illustrating PIEZO1-HaloTag CRISPR knock-in in WTC-11 hiPSCs; multiple human cell types differentiated from the PIEZO1-HaloTag hiPSC line; and subsequent HaloTag-ligand probe labeling and imaging. **B** Structural schematic of the trimeric PIEZO1 channel (Dark blue, PDB: 5Z10) with HaloTag (Cyan, PDB: 5UY1) attached to the cytosolic C-terminus. Dashed gray lines highlight the linker sequence (G-S-G-A-G-A) between PIEZO1 and HaloTag. **C** Representative traces of cell-attached patch clamp measurements with mechanical stimulation imparted through negative suction pulses for ECs derived from WTC-11, PIEZO1 KO, and PIEZO1-HaloTag hiPSCs. Blue color gradient indicates strength of negative pressure steps associated with suction pulses (light blue: lowest pressure, darkest blue: highest pressure). **D** Maximal suction-evoked current amplitudes recorded in each condition from 5 independent

experiments. All values are expressed as mean ± SEM (WTC-11 mean: − 71 ± 12.1 pA, $n = 21$; PIEZO1 KO mean: − 10.6 ± 7.5 pA, $n = 17$; PIEZO1-HaloTag mean: − 60.8 ± 13.4 pA, $n = 24$) ***$p$-value < 0.005, Mann-Whitney. WTC-11 compared to PIEZO1 KO $p = 8 \times 10^{-6}$; PIEZO1-HaloTag compared to PIEZO1 KO $p = 2 \times 10^{-4}$. Cohen's $d$ effect sizes are − 1.31 for PIEZO1 KO as compared to WTC-11. WTC-11 and PIEZO1-HaloTag did not show a statistically significant difference ($p$-value = 0.58) **E** TIRF images, representative of 3 independent experiments, showing unmodified WTC-11 hiPSCs (left), PIEZO1-HaloTag hiPSCs (middle), and PIEZO1-HaloTag KO hiPSCs (right), all treated with JF646 HTL. **F** TIRF images, representative of 3 independent experiments, of differentiated PIEZO1-HaloTag EC, keratinocyte, and NSC labeled with JF646 HTL. See also Supplementary Figs. 1, 2, and 3 and Supplementary Movies 1, 2, and 3. Data points are shown as mean ± SEM unless otherwise noted.

noise ratio. We compared the spatial and temporal resolution of data obtained from PIEZO1-tdTomato Liver Sinusoidal Endothelial Cells (mLSECs) harvested from PIEZO1-tdTomato mice and from PIEZO1-HaloTag hiPSC-derived ECs (Fig. 2A).

We labeled PIEZO1-HaloTag ECs with Janelia Fluor 549 HaloTag ligand (JF549 HTL), enabling consistent experimental settings; i.e., the same 561 nm laser wavelength and power settings, filters, and camera acquisition settings as for tdTomato. Under the imaging conditions used, the initial integrated intensity shown in camera units (c.u.) (see Methods) of JF549 HTL-labeled PIEZO1-HaloTag puncta (4179 ± 268.8 c.u.) was approximately twice that of the PIEZO1-tdTomato puncta (2420 ± 108.6 c.u.), and they bleached more slowly, (time constant $\tau = 38.1 ± 0.15$ s for PIEZO1-HaloTag vs. $\tau = 21.5 ± 0.13$ s for PIEZO1-tdTomato puncta (Fig. 2A). Here and throughout, all means are shown as ± SEM. The mean signal-to-background ratio was 5.62 ± 0.19 for PIEZO1-HaloTag puncta as compared to 3.14 ± 0.12 for PIEZO1-tdTomato puncta. Furthermore, the localization error, assessed by measuring frame-by-frame deviations from the mean position of puncta in fixed cells, was smaller for JF549 HTL-labeled PIEZO1-HaloTag channels (33 ± <1 nm) compared to fixed PIEZO1-tdTomato channels (52 ± <1 nm) (Fig. 2B). Within our fixed sample data, we found a negligible contribution of x-y drift (<10 nm over 5 s) (see Methods). Thus, PIEZO1-HaloTag puncta were brighter, bleached more slowly,

and enabled greater localization precision compared to PIEZO1-tdTomato puncta.

To illustrate the improved imaging capability of the PIEZO1-HaloTag system, we used TIRF microscopy to monitor a migrating PIEZO1-HaloTag NSC labeled with JF646 HTL (Fig. 2C and Supplementary Movie 4). We have previously shown that PIEZO1 is enriched at the rear of migratory cells; however, rapid photobleaching of the PIEZO1-tdTomato fluorophore precluded fast imaging of PIEZO1 dynamics during cell migration[13]. By imaging PIEZO1-HaloTag channels in a migrating NSC over 2 min at a temporal resolution of 10 frames per second (fps) (Fig. 2C and Supplementary Movie 4), we observed PIEZO1 channel enrichment at the trailing end of the cell throughout rear retraction, with a subset of PIEZO1 puncta displaying linear organization at the back of the cells. These observations highlight the enhanced imaging capacity of the PIEZO1-HaloTag system, allowing for high-resolution visualization of PIEZO1 dynamics and organization during cellular processes.

### Super-resolution tracking of PIEZO1 puncta reveals distinct mobility modes

To analyze the mobility of PIEZO1 in the membrane, we used the FIJI plugin ThunderSTORM[48,49] for super-resolution localization of PIEZO1-HaloTag puncta in TIRF image stacks. Successive localizations were

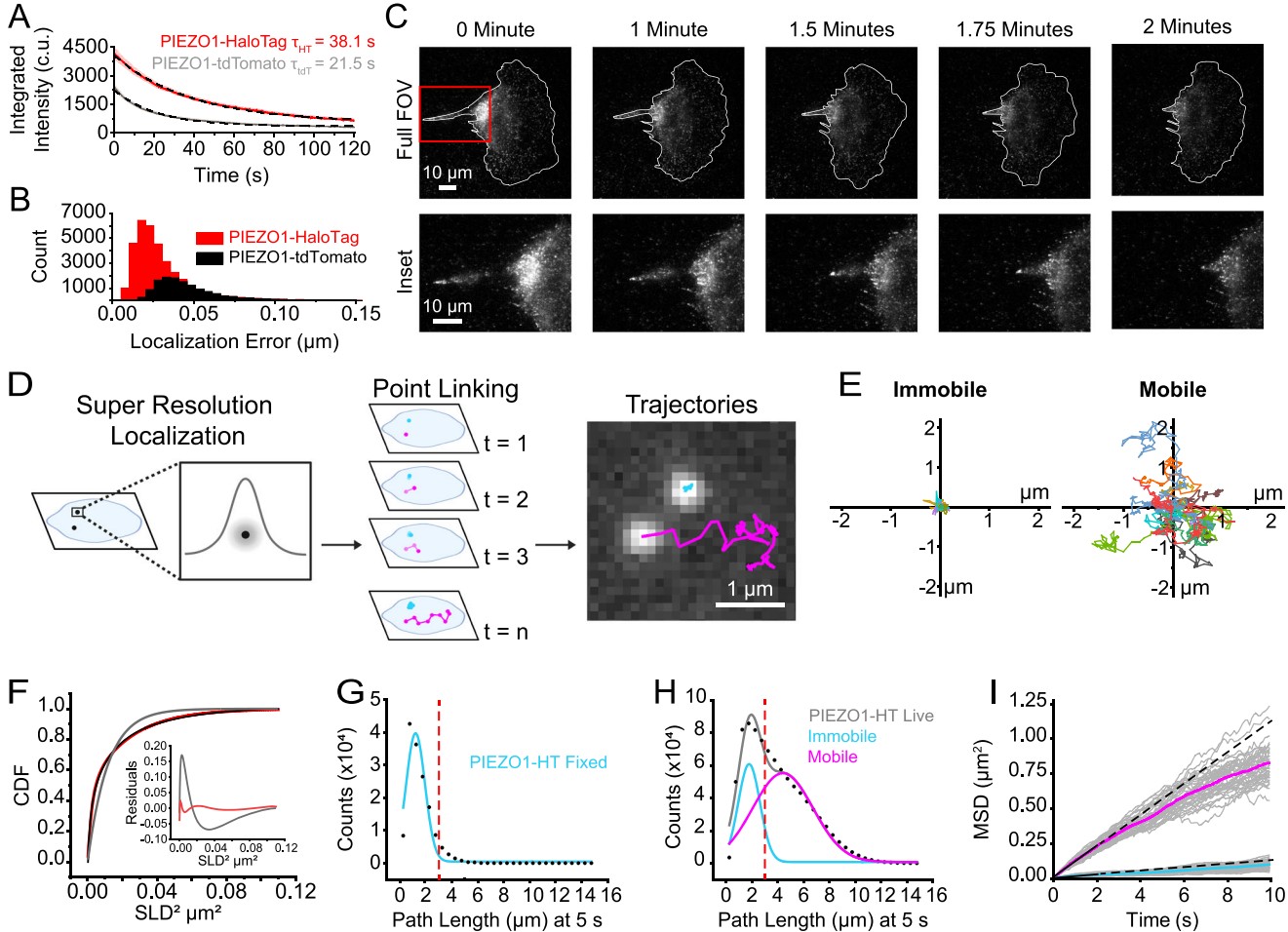

**Fig. 2 | PIEZO1-HaloTag localization and tracking reveal populations of PIEZO1 with different mobilities. A** Data from TIRF videos of PIEZO1-HaloTag ECs labeled with JF549 HTL and of PIEZO1-tdTomato mLSECs, acquired for 2 min with identical settings. Red trace indicates the average fluorescence intensity of PIEZO1-HaloTag puncta ($n = 19$ videos from 3 experiments), and gray trace for PIEZO1-tdTomato puncta ($n = 20$ videos from 3 experiments). Black dashed curves represent exponential fits, with $\tau_{HT} = 38.1 \pm 0.15$ s and $\tau_{tdT} = 21.5 \pm 0.13$ s (Mann-Whitney $p$-value $= 5.29 \times 10^{-7}$, Cohen's $d = -2.73$). **B** Puncta localization error distributions for JF549-HTL-labeled PIEZO1-HaloTag ECs and PIEZO1-tdTomato mouse fibroblasts (see Methods), imaged with identical settings. **C** TIRF images of a migrating JF646-HTL-labeled PIEZO1-HaloTag NSC, representative of 3 independent experiments, imaged at 10 fps. The top row shows the full cell, bottom row a zoomed-in region. **D** Schematic of super-resolution single particle tracking of PIEZO1-HaloTag puncta, with two mobility behaviors: mobile (magenta) and immobile (cyan). **E** Representative trajectories from 11 immobile and 11 mobile puncta tracked for 10 s, with starting positions normalized to the origin; color indicates different trajectories. **F** Cumulative distribution functions (CDF) of Single Lag Displacements (SLD). The black dotted curve is experimental data; gray and red curves represent single- and two-component exponential fits, respectively. The inset shows residuals for both fits. **G** Path length distribution at 5 s from PIEZO1-HaloTag ECs after fixation (14,943 trajectories, 13 videos). The dashed red line indicates the 3-μm cutoff for immobile trajectories. **H** Path length distribution for live JF646-HTL-labeled PIEZO1-HaloTag ECs (889,971 trajectories, 40 videos). The gray curve represents a fit to a sum of two Gaussian curves; individual curves shown in cyan and magenta. The dashed red line indicates the cutoff discriminating mobile and immobile trajectories. **I** Mean squared displacement (MSD) for immobile (cyan) and mobile (magenta) puncta. Gray traces represent mean MSD for each video; solid curves represent mean MSD across all videos; black lines are linear fits to data at $t < 2$ s, dashed black lines show linear extrapolations. Data for panels E-I are from 4 independent experiments. See also Supplementary Movie 4. Data points are shown as mean ± SEM.

linked to form trajectories using the custom-built, open-source image processing software FLIKA[50] (Fig. 2D and Supplementary Movies 4 and 5, see Methods). We compared the number of tracked puncta detected from PIEZO1-HaloTag KO ECs to PIEZO1-HaloTag ECs, both labeled with JF646 HaloTag ligand. PIEZO1-HaloTag cells yielded $0.5 \pm 0.05$ tracked puncta/μm² while knockout cells yielded $0.004 \pm 0.0006$ tracked puncta/μm² (***$p < 0.005$, Cohen's $d = -5.44$), indicating that our analysis methods yielded minimal spurious trajectories. Visual inspection of PIEZO1-HaloTag trajectories revealed the presence of two distinct populations of puncta: those that appeared immobile and others that were highly mobile (Fig. 2E and Supplementary Movie 4), concordant with recent findings by Ly et al[39]. Trajectories superimposed on a fluorescent image of the endoplasmic reticulum (ER) showed immobile and mobile puncta, both coincident

with the ER signal and in regions without ER (Supplementary Fig. 5). Thus, it appears that the mobility of puncta is not exclusively governed by intracellular organelles like the ER.

We calculated Single Lag Displacements (SLD), i.e., the distance covered by a punctum between consecutive frames, and generated a cumulative distribution function (CDF) of SLD² from the individual PIEZO1 trajectories (Fig. 2F). In a population with a single diffusive behavior, the CDF should fit a single exponential function. However, our data were not adequately fit by a single-component exponential, whereas a two-component exponential fit well (Fig. 2F), consistent with the presence of two distinct mobility behaviors. To segregate the two populations, we separated the tracks of individual puncta based on the total path length traveled within a period of 5 s. We first determined how much apparent movement would result from localization error by

labeling PIEZO1-HaloTag ECs with JF646 HTL and imaging them after fixing the samples. JF646 HTL labeled PIEZO1-HaloTag puncta had a localization error of $29 \pm <1$ nm. The apparent path lengths for these fixed puncta over 5 s (representing a summation of localization errors over 50 frames) followed a roughly Gaussian distribution, peaking at about 1.20 μm (Fig. 2G). In contrast, trajectories from live PIEZO1-HaloTag ECs labeled with the same probe showed a distribution of path lengths that was fitted well by a two-component Gaussian distribution (Fig. 2H). The first component of the Gaussian fit (peak 1.78 μm) approximated that of the fixed cell data, suggesting that this fraction of PIEZO1 puncta is indeed almost immobile in live cells. To extract the second, mobile component, we selected a cutoff value of path length of >3 μm at 5 s to largely exclude 'immobile' puncta (see dashed red lines in Fig. 2G and Fig. 2H). Figure 2I shows plots of the mean squared displacements (MSD) vs. time for classes of immobile and mobile puncta as segregated by this criterion. Traces for these populations separated into two distinct groups (Fig. 2I). Puncta undergoing Brownian (random) diffusion would display a straight line on this plot, with the diffusion coefficient $D = d^2/4t$, where d is the mean distance from the origin at time t. A linear fit up to 2 s for the mobile population yielded an apparent diffusion coefficient of 0.029 μm$^2$ s$^{-1}$, representing the upper limit of the diffusion coefficient. At longer times, the relationship fell below linear (Fig. 2I), indicating sub-Brownian, anomalous diffusion, similar to our recent observations with PIEZO1-tdTomato channels[39]. In contrast, the diffusion coefficient for the immobile population based on the linear fit up to 2 s yielded a maximal diffusion coefficient of 0.003 μm$^2$ s$^{-1}$. Using the same path length cutoff value in PIEZO1-HaloTag NSCs also yielded two distinct populations of mobile and immobile PIEZO1 puncta (Supplementary Fig. 6). The diffusion coefficients for mobile PIEZO1 puncta (0.029 μm$^2$ s$^{-1}$ in ECs and 0.026 μm$^2$ s$^{-1}$ in NSCs) fall within the expected diffusion range for large, membrane ion channels as reported diffusion coefficients for IP$_3$R (0.064 μm$^2$ s$^{-1}$)[51], TRAAK in a freestanding bilayer (0.8 μm$^2$ s$^{-1}$)[52], and PIEZO1 in red blood cells (0.037 μm$^2$ s$^{-1}$)[53] are similar to our reported values.

## Monitoring PIEZO1 activity with a Ca$^{2+}$-sensitive HaloTag ligand

The availability of Ca$^{2+}$-sensitive HTLs[44] enables measurement of channel activity using the PIEZO1-HaloTag system. We labeled PIEZO1-HaloTag ECs with Janelia Fluor 646-BAPTA HaloTag Ligand (JF646-BAPTA HTL), a non-ratiometric Ca$^{2+}$-sensitive fluorescent indicator[44]. The location of the HaloTag at the PIEZO1 C-terminus, near the cytosolic pore region of the channel (Fig. 1B), optimally places the probe to report on instantaneous Ca$^{2+}$ influx as increases in fluorescence intensity ("flickers").

To evaluate the Ca$^{2+}$ sensitivity of the probe, we transfected monomeric HaloTag into WTC-11 hiPSCs and labeled the cells with a 1:1 mixture of the Ca$^{2+}$-sensitive HTL JF646-BAPTA and the non-Ca$^{2+}$-sensitive JF549 (Supplementary Fig. 7). After cell fixation and permeabilization, we performed TIRF microscopy over a range of free Ca$^{2+}$ concentrations (0–39 μM). In the absence of Ca$^{2+}$, JF646-BAPTA-labeled puncta displayed barely detectable fluorescence, whereas JF549-labeled puncta showed stable fluorescence with little flickering (Supplementary Fig. 7). At a free Ca$^{2+}$ concentration of 75 nM, JF646-BAPTA-labeled puncta exhibited increased flickering to a bright state, and at 39 μM showed largely persistent bright fluorescence (Supplementary Fig. 7). Given the high local Ca$^{2+}$ concentration (>15 μM) expected close to the pore of an open channel[54], signals from JF646-BAPTA HTL PIEZO puncta would reflect channel gating, rather than random binding and unbinding of Ca$^{2+}$ ions to the HTL.

We next imaged live ECs labeled with JF646-BAPTA HTL in a bath solution containing 3 mM Ca$^{2+}$. We observed a relatively sparse density of detected puncta (0.07 ± 0.004 puncta per μm$^2$) as compared to cells labeled with the non Ca$^{2+}$sensitive JF646 HTL (0.34 ± 0.01 puncta per μm$^2$) (Fig. 3A). ECs derived from the PIEZO1-HaloTag KO hiPSCs had an even lower puncta density of 0.02 ± 0.002 per μm$^2$. Taken together, these observations suggest that a small fraction of PIEZO1 puncta are active in unstimulated cells. To verify that detected JF646-BAPTA HTL signals reflect PIEZO1 activity, we applied 2 μM Yoda1, a chemical agonist of the channel that increases the open-state occupancy of PIEZO1[28,55]. Incubating PIEZO1-HaloTag ECs with Yoda1 substantially increased the density of detected puncta to 0.22 ± 0.02 per μm$^2$ (Fig. 3A and B).

## Monitoring activity of endogenous human PIEZO1 with high temporal resolution

To visualize channel activity dynamics, we imaged PIEZO1-HaloTag ECs labeled with JF646-BAPTA HTL at 200 fps (Supplementary Movie 6). The fluorescence intensity profiles of PIEZO1-HaloTag JF646-BAPTA puncta showed flickers above the baseline (Supplementary Movie 7). We plotted puncta fluorescence intensity in the bright state across every frame for untreated cells, DMSO control, and 2 μM Yoda1-treated cells. The presence of Yoda1 resulted in both an increase in the number of bright-state puncta as well as an increase in their fluorescence intensity (untreated peak: 18.95 c.u., DMSO peak: 20.85 c.u., 2 μM Yoda1 peak: 22.75 c.u.) (Fig. 3C). To quantify fluctuations between the dim and bright state, we initially focused on immobile puncta for our analysis (Supplementary Movies 7–9). Flickers from individual puncta could be clearly resolved in fluorescence traces (Fig. 3D–G; Supplementary Movies 7, 9 and Supplementary Figs. 8, 9). All-points amplitude histograms showed the presence of multiple intensity levels (Fig. 3E, G, Supplementary Movie 7, 9 and Supplementary Figs. 8, 9). Between flickers, the signal at PIEZO1-HaloTag puncta sites was generally indistinguishable from the background fluorescence at surrounding regions, indicating that the intrinsic fluorescence of the BAPTA probe is very low at resting cytosolic [Ca$^{2+}$]. The absolute fluorescence values corresponding to the peaks of each flicker level varied between different puncta (Fig. 3D, F, and Supplementary Figs. 8–10) - possibly a result of different axial locations of the plasma membrane within the evanescent field, variable labeling efficiency of the HTL across puncta, or variations in puncta stoichiometry. Treatment with Yoda1 increased the proportion of time that puncta exhibited increased fluorescence and increased the proportion of time at higher intensity levels (Fig. 3F, G, Supplementary Movie 8 and Supplementary Fig. 10).

We next plotted an average all-points amplitude histogram from untreated, DMSO-treated, and 2 μM Yoda1-treated immobile trajectories (Supplementary Figs. 8–10). The average all-points amplitude histogram from 21 puncta per condition showed that PIEZO1-HaloTag fluorescence had a dim-state peak near 0 c.u. (background) across all conditions. The bright state intensity increased with Yoda1 treatment, consistent with its role in promoting PIEZO1 activation (Fig. 3H).

To further evaluate the temporal resolution of our recordings, we imaged JF646-BAPTA HTL signals in ECs at a faster frame rate of 500 fps (Fig. 3I). Transitions on both the rising and falling phases of flickers were complete within 2 frames (Fig. 3I), indicating that the fluorescence recordings track channel gating with a time resolution of 4 ms or better.

Next, we extended our analysis to mobile JF646-BAPTA HTL-labeled puncta, achieving simultaneous detection of channel location and activity (Fig. 3J and Supplementary Movie 9). However, a current limitation is that the dim basal fluorescence of the JF646-BAPTA HTL generally precluded the detection of a punctum when it was inactive, so tracks were restricted to durations when a punctum was in the bright state or could be interpolated across dim state gaps of a few frames (see Methods). Taken together with measurements from immobile puncta above, this demonstrates that both immobile and mobile PIEZO1 puncta can be active.

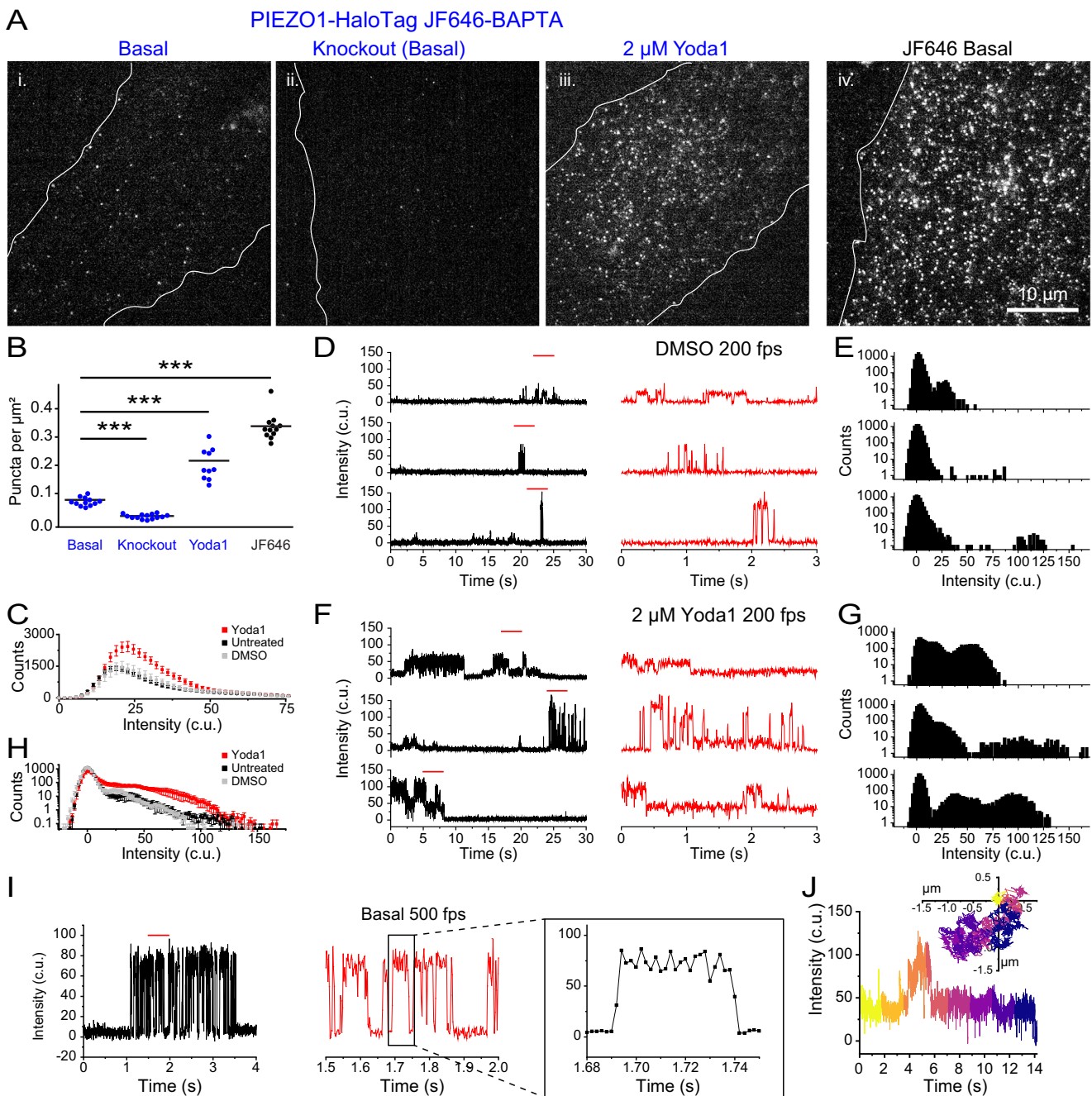

**Fig. 3 | PIEZO1-HaloTag enables imaging of endogenous PIEZO1 activity with temporal resolution approaching that of patch-clamp electrophysiology.**
**A** TIRF images of ECs labeled with either JF646-BAPTA HTL or JF646 (non-Ca²⁺-sensitive HTL). **B** Puncta densities (per μm²) for JF646-BAPTA HTL (blue) and JF646 (black) from images as in A. JF646-BAPTA HTL: Basal, 0.07 ± 0.004 (*n* = 12); PIEZO1-HaloTag KO, 0.02 ± 0.002 (*n* = 14); 2 μM Yoda1, 0.22 ± 0.02 (*n* = 11); JF646: 0.34 ± 0.01 (*n* = 12). Data from 4 independent experiments. Statistical comparisons: JF646-BAPTA HTL Basal vs. PIEZO1-HaloTag KO (two-sample *t* test *p* = 4.01 × 10⁻¹¹, Cohen's *d* = − 4.46), JF646-BAPTA HTL Basal vs. 2 μM Yoda1 (two-sample t-test *p* = 1.25 × 10⁻⁶, Cohen's *d* = 2.28) and JF646-BAPTA HTL Basal vs. JF646 (Mann-Whitney *p* = 3.62 × 10⁻⁵, Cohen's *d* = 7.72); ***p* < 0.005 for all conditions with. **C** Distributions of JF646-BAPTA puncta intensities for all puncta from 32 untreated, 12 DMSO, and 13 2 μM Yoda1 videos. **D** Representative background-subtracted fluorescence intensity traces of immobile PIEZO1-HaloTag puncta in TIRF imaging of JF646-BAPTA-labeled PIEZO1-HaloTag ECs treated with DMSO. Right, expanded traces corresponding to the red lines. **E** All-points amplitude histograms from

30 s traces in (**C**), shown on a log₁₀ scale. **F** Representative background-subtracted fluorescence intensity traces of immobile puncta from JF646-BAPTA-labeled PIEZO1-HaloTag ECs treated with 2 μM Yoda1. **G** All-points amplitude histogram of puncta intensities from the 30-s traces in (**F**). **H** Average of all-points amplitude histograms from 21 immobile puncta for untreated, DMSO, or 2 μM Yoda1-treated JF646-BAPTA-labeled PIEZO1-HaloTag ECs. Yoda1 distribution was significantly different with: Yoda1 vs. untreated *p* = 4.12 × 10⁻⁷, Yoda1 vs. DMSO *p* = 1.61 × 10⁻⁷, untreated vs. DMSO *p* = 0.05 (3 independent experiments, Two-Sample Kolmogorov-Smirnov test). **I** Representative fluorescence intensity trace of an immobile JF646-BAPTA-labeled PIEZO1-HaloTag punctum imaged at 500 fps. Expanded trace shown on the right. **J** Representative trajectory and fluorescence intensity trace of a mobile JF646-BAPTA-labeled PIEZO1-HaloTag punctum. Data for panels C-G are from 3 independent experiments. See Supplementary Figs. 8, 9, and 1,0 and Supplementary Movies 7, 8, and 9. Data points are shown as mean ± SEM unless otherwise noted.

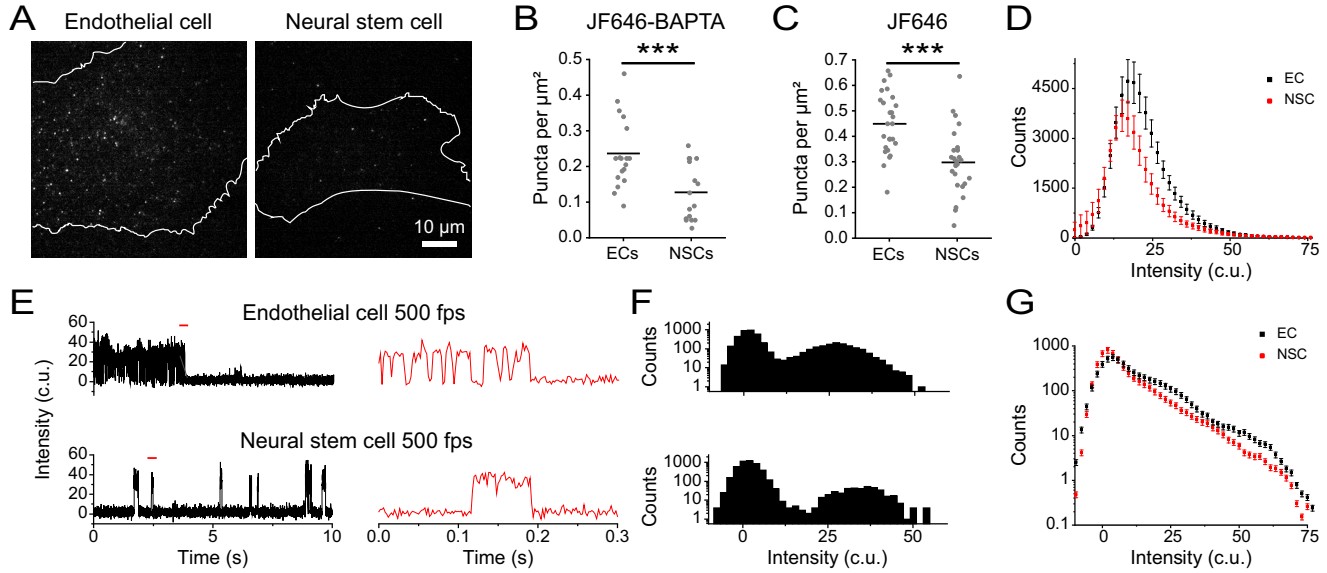

**Fig. 4 | PIEZO1 activity monitored in endothelial and neural stem cells.**
**A** Representative single frame TIRF images of PIEZO1-HaloTag ECs and NSCs labeled with JF646-BAPTA HTL. **B** JF646-BAPTA HTL puncta density in ECs (mean = 0.23 ± 0.02 puncta per µm², $n$ = 18) and NSCs (mean = 0.13 ± 0.02 puncta per µm², $n$ = 16) from 3 independent experiments. Each dot indicates the puncta density in a cell (see Methods). Means are indicated by black lines. The two groups were significantly different from one another, ***$p$-value = 0.003, Mann-Whitney. Cohen's $d$ effect size − 1.20. **C** Same as (**B**), for JF646 HTL puncta density in ECs (mean = 0.45 ± 0.02 puncta per µm², $n$ = 26 cells) and NSCs (mean = 0.30 ± 0.02 puncta per µm², $n$ = 26 cells) from 3 independent experiments, *** $p$-value = 7.37 × 10⁻⁵, Mann-Whitney. The Cohen's $d$ effect size was − 1.21. **D** Distributions of puncta intensities from all detected (i.e., bright state) JF646-BAPTA puncta across every frame in EC videos ($n$ = 15) and NSC videos ($n$ = 16).

Videos were recorded at 500 fps over 10 s, from 3 independent experiments. **E** Representative background-subtracted fluorescence intensity traces of immobile PIEZO1-HaloTag puncta from 500-fps TIRF imaging of an EC (top) and an NSC (bottom) labeled with JF646-BAPTA. *Right*, expanded traces from the sections marked with a red line on the left. **F** Corresponding all-points amplitude histograms from the 10-s traces shown in (**E**). Counts are shown on a log₁₀ scale. **G** Average of all-points amplitude histograms from 46 immobile puncta from 3 independent experiments of JF646-BAPTA labeled PIEZO1-HaloTag ECs (black) and NSCs (red). The two distributions were significantly different from each other, using a Two-Sample Kolmogorov-Smirnov test ($p$-value = 0.002). For individual fluorescence traces and all-points amplitude histogram of each punctum, see Supplementary Figs. 11, 12. Data points are shown as mean ± SEM unless otherwise noted.

## Investigating cell type-specific PIEZO1 activity

The functional roles of PIEZO1 in a given cell type may depend on the number and behavior of active channels specific to that particular lineage. To examine PIEZO1 behavior across two different cell types, we labeled ECs and NSCs with JF646-BAPTA HTL (Fig. 4A) and first quantified the density of puncta active in a frame (Fig. 4B). NSCs had close to half of the active puncta per unit area (mean = 0.13 ± 0.02 puncta per µm²) compared to ECs (mean = 0.23 ± 0.02 puncta per µm²). To assess whether cell-type-specific differences in PIEZO1 expression contribute to the lower density of active NSC PIEZO1-HaloTag puncta, we labeled PIEZO1-HaloTag ECs and NSCs with JF646 HTL. We noted that JF646 HTL-labeled NSCs had roughly 33% lower puncta density (mean = 0.30 ± 0.02 puncta per µm²) than ECs (mean = 0.45 ± 0.02 puncta per µm²) (Fig. 4C). However, this reduction in PIEZO1 expression was less pronounced than the nearly 50% decrease in PIEZO1 activity observed in NSCs compared to ECs, suggesting that our observed differences in active puncta density were at least partially, but not completely, due to different PIEZO1 expression levels in each cell type. We plotted the fluorescence intensity of JF646-BAPTA puncta in the bright state, showing that the peak intensity for ECs (17.05 c.u.) was higher than for NSCs (15.15 c.u.), with ECs also having more bright puncta at higher fluorescence intensity values (Fig. 4D). To better determine whether puncta in NSCs were less active, we imaged JF646-BAPTA-labeled cells at 500 fps and generated all-points amplitude histograms from these traces (Fig. 4E, F). We then plotted an average all-points amplitude histogram using 46 immobile puncta each from ECs and NSCs (Supplementary Figs. 11, 12), which showed an increased proportion of counts for EC PIEZO1 in the bright state relative to

NSC PIEZO1 (Fig. 4G). Therefore, PIEZO1-HaloTag imaging revealed differences across the two human cell types, with NSCs having fewer and less active puncta than ECs.

## PIEZO1-HaloTag signals report on mechanically-evoked activity of PIEZO1

To assess the effectiveness of the PIEZO1-HaloTag system in reporting PIEZO1's response to mechanical stimuli, we applied a hypotonic stimulus to PIEZO1-HaloTag ECs. The reduction in osmolarity is expected to increase membrane tension, thereby enhancing PIEZO1 activity[56–58]. To evaluate the PIEZO1-HaloTag response to osmotic shock, we labeled PIEZO1-HaloTag ECs with JF646-BAPTA HTL and imaged them in solutions of different osmotic strengths: 314 mOsm/L control; 251 mOsm/L hypotonic (Supplementary Fig. 13A). We observed a 29% increase in the density of bright JF646-BAPTA-labeled puncta following the hypotonic stimulus (Supplementary Fig. 13B), increasing from 0.28 ± 0.02 puncta per µm² in 314 mOsm/L to 0.36 ± 0.02 puncta per µm² in 251 mOsm/L. We then imaged JF646-BAPTA-labeled cells in control and hypotonic solution at 200 fps and tracked puncta to obtain their fluorescence intensity profiles and corresponding all-points amplitude histograms. Representative traces showed more channel activity in the hypotonic condition, which is also reflected in the all-points amplitude histogram (Supplementary Fig. 13C). We plotted an average all-points amplitude histogram of 71 tracked puncta in each condition (Supplementary Figs. 14, 15), which showed that the hypotonic stimulus increased the proportion of PIEZO1 channels in the open state compared to the control solution (Supplementary Fig. 13D). Thus, the PIEZO1-HaloTag system can be used to study PIEZO1's response to mechanical stress.

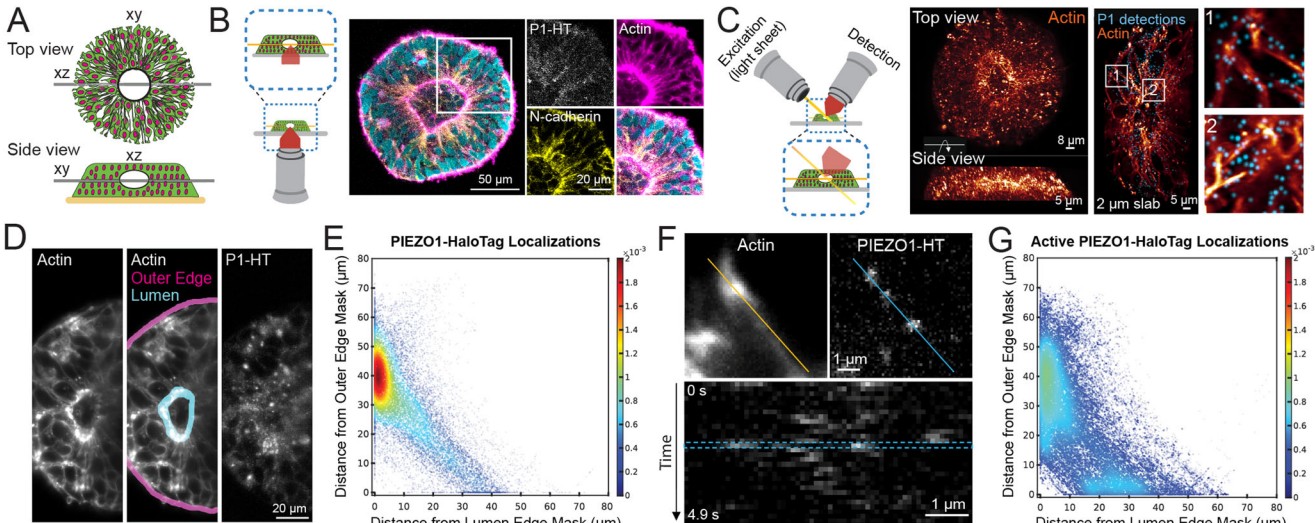

**Fig. 5 | Visualizing the spatial distribution and activity of PIEZO1-HaloTag puncta in micropatterned neural rosettes (MNRs). A** Top and side view schematics of an MNR, showing cells organized radially around a central lumen. **B** Left schematic shows the confocal imaging plane (orange) in an MNR. *Right*: Confocal image (representative of 3 independent experiments) of a PIEZO1-HaloTag MNR labeled with JF646 HTL, fixed, and stained with phalloidin (magenta), anti-N-cadherin (yellow), and Hoechst (cyan). Zoomed-in sections show PIEZO1-HaloTag at cell-cell interfaces and actin-rich regions at the lumen and outer edges. A gamma of 0.5 was applied to the zoomed-in actin image. **C** Left schematic showing lattice light-sheet microscopy imaging in MNRs. *Right*: Representative volumetric rendering of an MNR labeled with actin-phalloidin and JF635 HTL showing top-down and side views. *Middle*: A 2 μm slab projection of actin (orange) and PIEZO1-HaloTag detections (cyan) with zoomed-in insets on far right. **D** Representative maximum intensity projection (MIP) of 3 optical planes acquired 215 nm apart in an MNR. *Left*, actin channel, *middle*, lumen and outer edge masks, *right*, PIEZO1-HaloTag.

Representative images in C-D are from 4 independent experiments. **E** Density scatter plot of distances of PIEZO1-HaloTag puncta localizations to the lumen edge mask and outer edge mask ($n = 103$ videos from 21 MNRs, 4 independent experiments). The color scale indicates relative density of puncta at each position, scaled to the total number of puncta in the plot. The red cluster indicates PIEZO1 localization near the lumen edge. **F** Representative MIPs from 3-plane stacks of actin and Ca²⁺-sensitive JF646-BAPTA-labeled PIEZO1-HaloTag puncta in an MNR ($n = 3$ independent experiments). *Bottom*, kymograph generated from blue line (upper panel) showing fluorescence intensity flickering corresponding to PIEZO1 activity. **G** Density scatter plot of distances of active (bright) PIEZO1-HaloTag puncta localizations to the lumen edge mask and outer edge mask ($n = 39$ videos from 12 MNRs, 3 independent experiments). The color scale indicates relative density of puncta at each position, scaled to the total number of puncta in the plot. See also Supplementary Figs. 16–21, Supplementary Results, and Supplementary Movie 10.

## Imaging PIEZO1 in a tissue organoid model of neural development

The PIEZO1-HaloTag hiPSC line enables the investigation of PIEZO1 at the tissue scale, using organoid models derived from hiPSCs. To demonstrate this, we examined PIEZO1 localization and activity in Micropatterned Neural Rosettes (MNRs)[59,60], a hiPSC-derived human organoid model that mimics early neural development. MNRs exhibit a reproducible radial cell organization around a single central lumen, analogous to the cross-section of a neural tube (Fig. 5A). We generated MNRs from PIEZO1-HaloTag hiPSCs, labeled them with JF646 HTL, fixed the samples, and imaged them using confocal microscopy. Co-staining with SPY555-actin revealed an actin-rich lumen and outer edge, with radial actin signal between these two regions. Punctate HTL signal was located at cell-cell interfaces, and was more pronounced closer to the central lumen than at the MNR's outer edge (Fig. 5B).

Due to the limitations of confocal microscopy for volumetric time series imaging, we employed adaptive optical lattice light-sheet microscopy (AO-LLSM)[61] for rapid, high-sensitivity 3D imaging of live MNRs with minimal phototoxicity. We labeled live PIEZO1-HaloTag MNRs with JF635 HTL, as well as markers for actin and nuclei to delineate MNR morphology. We computationally separated the punctate PIEZO1 signals from the diffuse, non-specific auto-fluorescence also observed in PIEZO1-HaloTag KO MNRs and unlabeled PIEZO1-HaloTag MNRs (see Methods, Supplementary Results, and Supplementary Fig. 20). AO-LLSM imaging revealed the central lumen, the radiating actin cytoskeleton, and the PIEZO1-HaloTag JF635 HTL signal throughout the MNR volume (Fig. 5C and Supplementary Movie 10).

For quantitative analysis of PIEZO1-HaloTag puncta distribution in live MNRs, we captured images of 3 optical planes over time to generate a time series consisting of 30 time points for a total average duration of 5.5 s. We then generated a maximum intensity projection (MIP) image over the 3 planes to obtain an image stack over time. We manually constructed masks representing the lumen and outer edges of the MNR in the MIP images (Fig. 5D), computationally identified PIEZO1-HaloTag puncta over the duration of imaging, and measured the distances of each punctum from the lumen and the outer edge mask (see Methods, Supplementary Results). Density scatter plots of the distances showed a clustering of PIEZO1-HaloTag puncta near the lumen edge (Fig. 5E and Supplementary Fig. 16). For the entire dataset of 103 videos from 21 MNRs, PIEZO1-HaloTag puncta distances were distributed with modes close to 0 μm from the lumen edge mask and ~40 μm from the outer edge mask (Supplementary Fig. 16A), further supporting lumen enrichment. Cumulative distribution plots of distance data showed that $38 \pm 1\%$ of detected PIEZO1 puncta were within 5 μm of the lumen edge mask, while only $9 \pm 1\%$ were within 5 μm of the outer edge mask (Supplementary Figs. 8, 17A, 18A), indicating significant concentration of PIEZO1 channels at the lumen border.

To study PIEZO1 channel activity in MNRs, we labeled PIEZO1-HaloTag MNRs with the Ca²⁺-sensitive JF646-BAPTA HTL. Similar to TIRF measurements, we observed puncta exhibiting flickering behavior (Fig. 5F), albeit with a lower temporal resolution of the lightsheet modality. We computationally thresholded active PIEZO1-HaloTag puncta (see Methods, Supplementary Results, and Supplementary Fig. 21) and then quantified the distances of active PIEZO1-HaloTag puncta from the MNR lumen and outer edge masks, as for the JF635 HTL-labeled puncta above. We noted a diffuse cluster near the lumen edge and an additional smaller cluster near the outer edge (Fig. 5G and Supplementary Fig. 19). The localization of active PIEZO1 channels in

the vicinity of the actin-rich lumen and outer edge regions suggests that channels in these regions are activated by cell-generated acto-myosin forces, as we previously demonstrated in single cells[30]. Cumulative distribution plots of distance data from 39 videos in 12 MNRs revealed that $28 \pm 3\%$ of detected active PIEZO1 puncta localized within 5 µm of the lumen edge mask, while $20 \pm 5\%$ were within 5 µm of the outer edge mask (Supplementary Figs. 17B and 18B). Therefore, there were only 40% more active channels at the lumen border than near the outer edge, even though there were over four times as many channel puncta overall at the lumen, suggesting that tissue forces at the outer edge more efficiently activate PIEZO1 channels. These observations open future research avenues examining specific mechanisms by which channels are preferentially recruited to the lumen and outer edge regions, how cell- and tissue-level forces in MNRs activate PIEZO1, and the physiological impacts of this activity.

## Discussion

Here, we genetically engineered a human induced pluripotent stem cell (hiPSC) line by fusing a HaloTag protein to endogenous PIEZO1. By editing both alleles of the PIEZO1 gene, we ensured that all expressed PIEZO1 protein is tagged with the HaloTag. This modification, which preserves channel function, creates a chemogenetic tag for the channel that is compatible with a diverse array of specially designed Halo-Tag ligands (HTLs). We focus on imaging-based applications, utilizing the array of bright and photostable Janelia Fluor HTLs to visualize the localization and activity of individual PIEZO1 channels in a range of hiPSC-derived cell types and tissue organoids. Additional applications of the HaloTag technology, such as biochemical studies to identify interacting partners[62], high-yield protein purification[46], and Protac-mediated targeted protein degradation[63], are also possible. Our PIEZO1-HaloTag hiPSC system thus synergizes the multifunctionality of HaloTag technologies with the versatility of hiPSCs, offering an advantage over traditional methods in studying PIEZO1, a channel that has emerged as a critically important mechanotransducer in a wide range of physiological processes.

The PIEZO1-HaloTag hiPSC line allows for the visualization of individual mechanically-activated PIEZO1 puncta in the native cellular environment. This approach circumvents the limitations of over-expression systems, such as altered channel localization and activity due to changes in channel density. Our model also offers the flexibility to differentiate into multiple cell types and tissue organoids, providing a comprehensive understanding of PIEZO1 dynamics across spatial scales and physiological systems. The superior brightness and photo-stability of the Janelia Fluor HTLs allow us to capture PIEZO1 localization dynamics with higher precision than previous PIEZO1-tdTomato reporter models[13,30,39], revealing two distinct mobility behaviors and enabling imaging of the rear-enrichment of PIEZO1 in migrating cells with greater temporal resolution than previously possible[13]. Thus, our tool paves the way for a new generation of studies to examine PIEZO1 mobility and localization under a variety of physiological conditions.

A further outcome of our study is the establishment of a single-channel PIEZO1 activity assay within the native cellular milieu. The HaloTag domain is attached to the PIEZO1 C-terminus, located directly below the pore domain; thus, attachment of a $Ca^{2+}$-sensitive HTL enables PIEZO1 activity measurements by detecting $Ca^{2+}$ influx through the pore. This assay marks a significant advance over patch clamp electrophysiology, the standard method for measuring PIEZO1 activity. Whole-cell patch clamp dialyzes the cell, disrupting cellular structures and physiology, while cell-attached patch clamp imparts a large resting tension on the patch that may alter channel behavior. Neither modality provides spatial information, whereas imaging approaches provide multiplexed activity measurements from dozens to hundreds of channel puncta simultaneously while maintaining cell integrity[64]. We present several technical advances over previous imaging-based methods to measure PIEZO1 activity[26–28,30,32]. The PIEZO1-HaloTag

platform provides highly specific labeling of endogenous PIEZO1, single-channel measurement capabilities in intact cells, readouts of both mobility and activity, high precision spatial information PIEZO1 localization and activity, and high temporal resolution – making our tool well suited for examining PIEZO1 dynamics under native cellular conditions in both isolated cells and tissue organoids. Together, these advances open up new avenues to study PIEZO1 function and regulation in human systems.

Our findings build on existing methodologies, such as GenEPi[65], a genetically encoded fluorescent reporter of PIEZO1 activity, which relies on overexpression and is limited by poor kinetics and weaker signals. Activity measurements from PIEZO1-HaloTag revealed multiple levels of JF646-BAPTA HTL fluorescence intensity from single, stationary puncta. Although saturated labeling by the probe involves three HaloTag ligands per PIEZO1 trimer, the high $[Ca^{2+}]$ in the immediate proximity of a channel pore, estimated at $> 15\,\mu M$[54], implies that the three HTLs would respond almost simultaneously to $Ca^{2+}$ flux, given their proximity to the channel pore and high affinity for $Ca^{2+}$ (Kd $0.14\,\mu M$)[44]. It may also be possible that the observed "bright state" durations may overestimate channel open times as a consequence of the time courses of $Ca^{2+}$ unbinding from the HTL and of $Ca^{2+}$ diffusion away from the pore. We speculate that multiple levels of JF646 BAPTA HTL fluorescence may result if each diffraction-limited punctum represents a cluster of two or more PIEZO1 channels. Differences in amplitudes of signal from different PIEZO1-HaloTag puncta could thus arise from differences in the number of PIEZO1 channels per punctum, from varying distances of the membrane within the exponentially decaying evanescent field, or from sub-saturation labeling of some channels. Importantly, recent super-resolution studies using MINFLUX[56,66] imaging of PIEZO1 demonstrate clustering of PIEZO1[67].

Our studies also reveal the capability of the $Ca^{2+}$-sensitive JF646-BAPTA HTL to spatially map PIEZO1 mobility in addition to activity, and we find that both stationary and mobile puncta can be active. However, the low resting fluorescence of the HTL in the absence of bound $Ca^{2+}$ poses challenges in detecting and localizing PIEZO1-HaloTag channels when they are closed. This limitation may be addressed by the development of $Ca^{2+}$-sensitive HTL variants with higher resting-state brightness, and by further engineering the hiPSC line to enable orthogonal dual labeling of PIEZO1 with spectrally distinct $Ca^{2+}$-sensitive and insensitive HTLs.

The PIEZO1-HaloTag hiPSCs also allows for the comparison of relative differences in PIEZO1 activity between distinct cell types. NSCs display less active PIEZO1 puncta than ECs which we speculate may be due to differences in cell type-specific contractility that activates PIEZO1[30]. Using osmotic stimulis, we also provide proof-of-principle that the PIEZO1-HaloTag platform allows measurement of PIEZO1 activity in response to externally-applied mechanical stimuli. These activity measurements can be extended to other mechanical stimulus modalities in the future.

Beyond the single-cell level, our PIEZO1-HaloTag hiPSCs provide a platform for studying PIEZO1 at the tissue scale using in vitro human organoid models. Building upon prior research in a mouse model that reveals PIEZO1's function in neural development[12], we examined the channel's localization and activity in MNRs, an in vitro model of early human neural tube development. Utilizing AO-LLSM for high-resolution live imaging, we observed a notable enrichment of PIEZO1 channels at the lumen edge of MNRs. Interestingly, channel activity was observed at both the lumen and outer edges, areas marked by high mechanical tension as evidenced by strong actin staining. Previous measurements in micropatterned neuroectoderm models showed that the outer edge experiences high cell-generated forces[68], which are known to activate PIEZO1[30]. Our observations highlight the need to consider the site-specific biology of tissues and their unique geometrical and mechanical properties when studying the physiological roles of PIEZO1.

In conclusion, our study establishes a hiPSC-based approach to study endogenous PIEZO1. The adaptability of the hiPSC model, capable of differentiating into a variety of human tissue organoids, creates opportunities for mechanistic studies on PIEZO1 function at the tissue scale across different organ systems. In addition, pathogenic PIEZO1 Gain-of-Function[15,18,19] or Loss-of-Function[69] mutations can be introduced in the PIEZO1-HaloTag hiPSCs for disease modeling of human PIEZO1 channelopathies. Finally, the precision in measuring activity and localization of wild-type and mutant PIEZO1 channels provides opportunities for drug screening aimed at therapeutic interventions targeting PIEZO1. Our approach provides a platform for enhancing the understanding of PIEZO1's role in cellular mechanics and signaling in normal physiology as well as disease conditions, and holds significant potential for developing therapeutic strategies in PIEZO1-mediated human diseases. Future improvements, including continued developments in the rapidly evolving areas of HaloTag technologies as well as advanced imaging techniques, will further refine our ability to study PIEZO1's intricate role in human health and disease. More broadly, our approach could be applied to other $Ca^{2+}$-permeable membrane proteins.

## Methods

### Ethical statement

All human stem cell experiments were carried out in accordance with the guidelines of the Human Stem Cell Research Oversight (hSCRO) Committee of the University of California, Irvine. Stem cell lines used had no patient identifiers. Animal studies were carried out in accordance with approved Institutional Animal Care and Use Committee (IACUC) protocols of the University of California, Irvine.

### hiPSC culturing and maintenance

hiPSCs were maintained in mTeSR Plus Basal Medium (Cat. No.100-11300, STEMCELL Technologies) with Primocin (Cat. No.NC9392943, Invivogen) in an incubator at 37 °C with 5% $CO_2$. Cells were cultured on plates incubated with 10 μg/mL Vitronectin XF (Cat. No.07180, STEMCELL Technologies) for at least 1 h at room temperature prior to passaging. Media was changed every other day, and cells were passaged mechanically or as single cells (10,000 cells/cm²) every 4–7 days. For generating single cell suspensions of hiPSCs, cells were dissociated with Accutase (Cat. No.# 07920_C, STEMCELL Technologies) for 3–5 min in a 37 °C incubator. Accutase was diluted 1:1 with mTeSR Plus, 100 μg/ml Primocin, and 10 μM Y-27632 (Cat. No.SM-0013-0010, Biological Industries, USA), and cells were centrifuged at 210 × g for 5 min. For TIRF imaging, cells were seeded at 13,000 cell/cm² density on Vitronectin XF-coated MatTek dishes (Cat. No.P35G-1.5-14-C, MatTek Corporation). Cells were regularly checked for mycoplasma contamination and karyotypic abnormalities. The reagent list is also summarized in Supplementary Table 3.

### CRISPR engineering

All *PIEZO1* genetic edits were outsourced to Sythego, Menlo Park, CA and made in WTC-11 hiPSCs (Cat. No. GM25256, Coriell) with normal karyotype. The ICE (Inference of CRISPR Edits) software analysis package developed by Synthego was used for analysis of CRISPR editing data. All clones had normal karyotype and passed pluripotency tests (OCT4 and SSEA immunostaining) and lacked mycoplasma contamination after CRISPR editing. Karyotyping of clones was performed with Karystat analysis (ThermoFisher).

**PIEZO1-HaloTag hiPSCs.** Briefly, the strategy employed for the CRISPR engineering was as follows. hiPSCs were transfected with Cas9, sgRNA (5'-GUGGACUCGUGAGAAGGAGU-3') and a knock-in template sequence which was designed with a 5' homology arm (HA) of 496 bp upstream of the stop codon of PIEZO1, an 18 bp linker, the HaloTag coding sequence followed by the 3' HA containing the TAG stop codon

and 499 bp of the 3' untranslated region of PIEZO1. The 18 bp linker, ggatccggtgcaggcgcc, encodes the amino acid sequence GSGAGA. HaloTag 7 is 297 amino acids encoded by a 891 bp sequence. Genetically engineered cells were screened for the presence of the insertion by PCR of the genomic DNA with FWD primer (5'-3'): GCCAAGCTCATCTTCCTCTAC and REV primer (5'-3'): GAACAT GAAGGACTTGGTGAGTA, which should yield a 1669 bp product, while unedited gDNA would yield a 760 bp product.

**PIEZO1-HaloTag KO hiPSCs.** PIEZO1-HaloTag hiPSC cells were edited to obtain multiple homozygous indels within the PIEZO1 coding region in exons 6 and 43, and clonal lines were isolated. Cas9 and guide RNAs were transfected sequentially into PIEZO1-HaloTag hiPSC cells. Guide RNAs used were sgRNA (5'-UGGAUGCCAGCCCGACGGCA-3') targeting exon 6 and sgRNA (5'-UCCGCCUACCAGAUCCGCUG-3') targeting exon 43 of Piezo1. Forward primer 5'-AGGTAGACACTGGAGAGGGC-3' and reverse primer 5'-CAGAGGAGCAGCTGTGGATG-3' were used in PCR amplification of genomic DNA from transfected cells. Sequencing of the PCR fragment from Clone E1 revealed a homozygous -1 indel in exon 6. Indels in exon 43 were identified using the forward primer 5'-ACCTTCTCTGTCTCTCGGCT-3' and the reverse primer 5'-ACCTTCTCTGTCTCTCGGCT-3' for PCR amplification. Sequencing of the fragment revealed a homozygous a -5 indel in exon 43.

### hiPSC differentiation

**Neural Stem Cell (NSC) differentiation.** NSCs were differentiated from hiPSCs using STEMdiff Neural Induction Medium (Cat. No. 05839, STEMCell Technologies) monolayer culture protocol as per the manufacturer's instructions. Briefly, hiPSCs were passaged using Accutase as described above and resuspended in STEMdiff Neural Induction Medium containing SMADi and 10 μM Y-27632. Cells were plated at 2 × 10⁵ cells/cm² onto tissue culture plates coated with 10 μg/ml of CellAdhere Laminin-521 (Cat. No.77003, STEMCELL Technologies). Media changes (without Y-27632) were performed daily, and cells were passaged at days 7, 14, and 21. NSCs were used from Day 21 and cultured in STEMDIFF Neural Progenitor Medium (Cat. No. 05833, STEMCell Technologies).

**Endothelial Cell (EC) differentiation.** ECs were differentiated following the S1-S2 method[70]. hiPSCs were dissociated into single cells using Accutase and plated on 10 μg/mL Vitronectin XF (Cat. No. 07180, STEMCELL Technologies)-coated plates with 10 μM of Y27632 (Cat. No.SM-0013-0010, Biological Industries, USA) in mTeSR Plus Basal Medium (Cat. No.100-11300) with Primocin (Cat. No.NC9392943, Invivogen). 3 × 10⁵ cells were seeded per well of a 6-well tissue culture plate. 24 h after seeding, the media was changed to S1 media. S1 media was prepared first by making a Basal medium consisting of Advanced Dulbecco's modified Eagle's medium (DMEM)/F12 (Cat. No. 12334010, Thermo Fisher Scientific), 1x Glutamax supplement (Cat. No. 35050061, Thermo Fisher Scientific), 60 μg/ mL L-Ascorbic Acid (Cat. No. A8960, Sigma- Aldrich), and Primocin. Basal media was supplemented with 6 μM CHIR99021 (Cat. No. SML1046-5MG, Sigma-Aldrich) to create S1 media. The media was changed every 24 h for two days. After two days in S1 media, the media was changed to S2. S2 media was prepared by supplementing Basal media with 10 μM SB431542 (Cat. No. S1067, Selleck Chem), 50 ng/mL bFGF-2 (Cat. No. 100-18B-100ug, PeproTech), 50 ng/mL VEGF-A (Cat. No. 100-20, PeproTech), and 10 ng/mL EGF (Cat. No. AF-100-15-100ug, PeproTech). The media was changed every 24 h throughout the protocol. After a total of 48 h in S2 media, cells were purified using an immunomagnetic CD31 MACs sorting kit (Cat. No. 130-091-935, Miltenyi) and 15,000 cells were plated on 10 μg/mL Fibronectin (Cat. No. 356008, Corning)-coated MatTek dishes (Cat. No. P35G-1.5-14-C, MatTek Corporation) and cultured for 48 h in EGM-2 media (Cat. No.CC-3162, Lonza) before imaging.

**Keratinocyte differentiation.** Mechanically passaged hiPSCS were plated on 10 µg/mL Vitronectin XF in 6-well TC plates in 1 µM all-trans-RA (Cat. No. R2500-25MG, Sigma-Aldrich) to NutriStem hPSC XF Medium (Growth Factor-Free)(Cat. No.06-5100-01-1 A, Biological Industries Israel Beit-Haemek Ltd.). Daily media changes were performed with media containing 1 µM RA for 7 days. On day 7, differentiated cells were passaged with 2 mg/mL of Dispase (Cat. No. CnT-DNP-10, Cellntec) incubated for 30 min at 37 °C. Cells were resuspended in 1 mL of media CnT-Pr media (Cat. No. CNT-PR, CellnTec) and 1/10th of total cells were plated on 10 µg/mL Fibronectin (Cat. No. 356008, Corning) coated MatTek dishes (Cat. No.P35G-1.5-20-C, MatTek Corporation). Cells were cultured in CnT-Prime Epithelial Proliferation Medium (Cat. No. CNT-PR, CellnTec) for 2–4 weeks prior to imaging. Media was exchanged to CnT-Prime Epithelial 2D Differentiation Medium (Cat. No. CnT-PR-D, CellnTec) for 2–4 days prior to imaging.

**Preparation and differentiation of Micropatterned Neural Rosettes (MNRs).** MNRs were generated from hiPSCs using a method developed by Haremaki et al.[60] using Arena A CYTOOchips (Cat. No.10-020-00-18, CYTOO INC.). A CYTOO Arena A chip was placed on parafilm in a 10-cm petri dish (Cat. No. 08757100D, Fisher Scientific), the top side was coated with 10 µg/ml of CellAdhere Laminin-521 (Cat. No.77003, STEMCELL Technologies) for 3 h at 37 °C. Laminin was then removed via multiple washes in PBS + calcium + magnesium (PBS +/+) (Cat. No. PBL02-500ML, Caisson Laboratories Inc.). The CYTOOchip was then moved to a single well of a 6-well tissue culture plate with PBS +/+ prior to plating. PIEZO1-HaloTag and PIEZO1-HaloTag KO hiPSCs were washed with PBS +/+ and incubated with accutase at 37 °C for 3–5 min to generate a single cell suspension. Accutase was diluted 1:1 with mTeSR Plus, containing 10 µM Y-27632, and spun at 210 g for 5 min. Cells were resuspended, counted, and seeded at a concentration of $5 \times 10^5$ hiPSCs in 2 mL of media on top of the micropattern in a single well of a 6-well plate. The media was changed after 3 h to differentiation media comprising of a 1:1 mixture of DMEM/F12 and Neurobasal containing 1:100 Glutamax, 1:200 Non Essential Amino Acids, 1:200 N2 supplement, 1:100 B27 without vitamin A (all from Invitrogen), 3.5 µL l-1 2-mercaptoethanol (Sigma), 1:4000 insulin (Sigma), 10 µM SB431542 (Cat. No. S1067, Selleck Chem) and 0.2 µM LDN193189 (Cat. No. SM-0005-0010, Biological Industries, USA). Media was changed daily, and MNRs were kept in culture until day 5 in a 37 °C incubator, at which time central lumens were present. MNRs were incubated with HaloTag Ligand as detailed below and imaged. To label actin structures, MNRs were incubated with SPY555-actin (Cat. No. CY-SC202, Cytoskeleton) at 1:1000 in culture media, and 1:1000 SPY505-DNA (Cat. No. CY-SC101, Cytoskeleton) was added to label DNA, both for 1 h prior, to imaging. MNRs were washed 3 times in culture media prior to imaging.

Small adjustments were made to the above protocol to prepare MNRs for confocal microscopy. After labeling with HaloTag Ligand on Day 5, the CYTOO chips were fixed in a 4% v/v paraformaldehyde solution, containing 5 mM MgCl₂, 10 mM EGTA, and 40 mg/mL sucrose in PBS (pH = 7.3) for 10 min at room temperature and the chip placed in the bottom of a 35 mm glass-bottom dish without a cover glass (Cat. No. D35-14, Cellvis). Thus, the micropatterned surface was positioned at the bottom in the hole. Subsequently, biosafe glue (KWIK-CAST silicon sealant, World Precision Instruments) was added to the bottom of the 35 mm dish around the edges of the CYTOOchip to seal it in place. The chip was rehydrated with 2 mL PBS +/+ for 5 min, immunostained with anti-N-cadherin antibody (Cat. No. 6109920, BD Biosciences), actin was labeled with Phalloidin (1:40 dilution, Cat. No. A12379, Invitrogen) and nuclei with Hoechst (3 mg/ml Cat. No. H1399, Invitrogen) and then imaged on an Olympus Fluoview FV3000 confocal.

## Mouse Liver Sinusoidal Endothelial Cell (mLSEC) and fibroblast isolation and culture

Livers from the PIEZO1-tdTomato mice[3] were dissected and placed on a petri dish and minced with scalpel blades. Once the liver was minced, it was resuspended in a solution to further dissociate the tissue. This solution contained 1 mL of 2.5 units/mL dispase, 9 mL 0.1% collagenase II, 1 µM MgCl₂, and 1 µM CaCl₂ in Hanks Buffer. The tissue was incubated with the dissociation mixture for 50 mins in a tube rotator with continuous agitation at 37 °C. After the completion of the dissociation process, the dissociated tissue was filtered using 70 and 40 µm cell strainers. The dissociated cells were then washed twice in PEB buffer containing phosphate-buffered saline solution (PBS), 0.5% BSA, pH 7.2, EDTA 2 mM, and EDTA 2 mM. Once the pellets were washed, cells were placed in 1 mL PEB buffer and 30 µL CD146 microbeads (Cat. No. 130-092-007, Miltenyi Biotech) at 4 °C for 15 min under continuous agitation. An LS column (Cat. No. 130-042-401, Miltenyi Biotech) was primed with PEB buffer during this incubation. Once the mLSECs were selected for using the microbeads, the solution was then placed in the primed LS column to separate the mLSECs (Miltenyi Biotech). After the cells passed through the column, the column was washed 3 times with 5 mL PEB buffer. Any CD146-positive cells were eluted from the column using 5 mL warmed EGM-2 growth medium supplemented with EGM-2 bullet kit (Lonza). The cells were pelleted with a 300 g spin for 5 min and counted in 1 mL EGM-2 media. Glass-bottom #1.5 dishes were prepared by coating with 10 µg/mL Fibronectin (Cat. No. 356008, Corning). After performing the cell count, approximately 30,000 cells were plated on each fibronectin-coated glass-bottom dish. After 2 h, a media change was performed, and then the media changed after 48 h and samples were imaged 72 h after seeding. Fibroblasts used in Fig. 2B were harvested and cultured as described in Ly et al.[39].

## Cell lysis and immunoblotting

iPSC cells were lysed in RIPA buffer (Thermo Scientific 89901) containing 1 µM Dithiothreitol, Thermo Scientific Halt Protease Inhibitor Cocktail (Cat. No. 78430), and Halt Phosphatase Inhibitor Cocktail (Part No. 78420)) for 15 min on ice. Lysates were sonicated (SONICS Vibra-cell, probe model CV184) at 100% power twice (10 s pulse on, 20 s off), then centrifuged 10 min × 16,100 g at 4 °C. The supernatant was transferred to a new tube, and the protein concentration was determined with Pierce BCA assays (Thermo Scientific 23225). Proteins were separated by SDS-PAGE by loading 15 micrograms of lysate on a NuPAGE 3–8% Tris Acetate gel, (ThermoFisher Cat. No. EA0375) in Tris Acetate SDS Running Buffer (Cat. No. LA0041), then transferred to 0.45 µm PDVF membrane (Thermo,FisherCat. No. 88518) with the Biorad Mini-PROTEAN Tetra System in 25 mM Tris, 192 mM Glycine, 20% Methanol overnight at 4 °C for 18 h at 30 V. Western blotting was performed with 1:1000 mouse anti-HaloTag (Promega, Cat. No. G9211) or 1:1000 mouse anti-PIEZO1 (Novus Biologicals, Cat. No. NBP2-75617), followed by 1:1000 Goat anti-Mouse IgG-Secondary antibody, HRP (Invitrogen, Cat No. 32430) or 1:100,000 HRP-mouse anti-actin (ThermoFisher Cat. No. 5-15739-HRP) in TBS, 0.1% Tween20 in 5% NonFat Dried Milk (Carnation). Antigen detection was performed with Thermo SuperSignal West Femto Max sensitivity (ThermoFisher, Cat. No. 34095) with a Bio-Rad Gel Chemi-Doc Molecular Imager. Reagents also summarized in Supplementary Table 1.

## HaloTag ligand treatment protocol

After differentiation of the PIEZO1-HaloTag hiPSC lines into either NSCs, ECs, Keratinocytes, or MNRs, each respective cell type was incubated at 37 °C with 500 pM of Janelia Fluor 646 HaloTag Ligand (NSCs, ECs, or keratinocytes) (Cat. No. GA1120, Promega) or Janelia Fluor 635 HaloTag Ligand (MNRs) (Cat. No. CS315103, Promega) or Janelia Fluor 646-BAPTA-3'AM- HaloTag ligand (requested from Lavis lab, Janelia Research Campus, HHMI) for incubation time outlined in

Supplementary Table 2. Upon testing a series of labeling conditions, we found that labeling overnight with a low concentration of Janelia Fluor 646 or Janelia Fluor 635 yielded clean labeling with low non-specific binding, likely due to the low expression level of endogenous PIEZO1 and high fluorogenicity of the JF-based probes[45,71]. Labeling conditions are provided in Supplementary Table 2. MNRs were treated with JF635 HTL due to its improved fluorogenicity and compatibility with existing filter sets in the AO-LLSM system[71]. Incubations with HTLs were performed in each respective cell type's basal media, outlined in their respective culturing sections. Cells were gently washed 5 times with DMEM/ F12 1:1 (Cat. No. 25116001, Invitrogen) at room temperature prior to imaging using TIRF microscopy. MNRs were washed 3 times with culture media prior to fixation or live imaging.

## Immunofluorescence

Samples plated in MatTek dishes (Cat. No. P35G-1.5-14-C or P35G-1.5-20-C, MatTek Corporation) were first fixed in a 4% v/v paraformaldehyde solution, including 5 mM $MgCl_2$, 10 mM EGTA, and 40 mg/mL sucrose in PBS (pH = 7.3) for 10 min at room temperature. Next, the fixed cells or MNRs were permeabilized with 0.3% Triton X-100 in PBS for 5 min. 5% BSA in PBS (Jackson ImmunoResearch, Cat. 001-000-162) for 1 h at room temperature was used to block nonspecific binding of excess antibodies. After blocking was complete, the sample was incubated overnight at 4 °C with primary antibody diluted in 1% BSA (Primary antibodies used and their concentrations are included in Supplementary Table 1). Next, samples were washed several times to remove the primary antibody and then were incubated with secondary antibody for 1 h at room temperature (Secondary antibodies used and their concentrations are included in Supplementary Table 1). Samples were washed several times and subsequently labeled with Hoechst (Cat. No. H1399, Invitrogen) at 1 µg/mL for 5 min at room temperature. Samples were stored in 1 × PBS prior to imaging.

## Cell-attached patch clamp

Cell-attached patch clamp experiments were made using an Axopatch 200B amplifier (Molecular Devices) at room temperature. Pipettes were made from thin-walled borosilicate glass capillaries (Warner Instruments) and contained a solution consisting (in mM) of 130 NaCl,10 HEPES, 10 tetraethylammonium chloride, 8 Glucose, 5 KCl, 1 $CaCl_2$, 1 $MgCl_2$, (pH 7.3) with NaOH. Bath solution composition included (in mM) 140 KCl, 10 Glucose, 10 HEPES, 1 $MgCl_2$, (pH 7.3 with KOH) was used to zero the membrane potential. The pipettes had a resistance of 0.6-1.1 MΩ in these solutions. Negative-suction application provided mechanical stimulation during recordings using a high-speed pressure clamp (HSPC-1; ALA Scientific) controlled using Clampex software. Suction pulses were applied using the patch pipette, and the membrane potential was held at −80 mV. Offline leak subtraction was performed prior to the quantification of maximum current. Maximum current was manually recorded and quantified for statistical differences using a two-sample t-test and mean effect size using Cohen's d.

## Whole-cell patch clamp

Whole-cell patch clamp experiments were conducted according to the protocol described previously with slight modifications[55]. Patch pipettes were pulled from thin-wall borosilicate capillaries with internal filament (GC 150TF-7.5; Harvard Apparatus). Recording pipettes were pulled to a resistance of 2–3 MΩ and fire polished using a microforge (MF2, Narishige). Standard HBSS (Cat. No. 14025092, GIBCO) was used as the bath saline. Patch pipettes were filled with 140 mM KCl, 10 mM HEPES, 10 mM TEA, and 2 mM EGTA (pH 7.4 with NaOH) saline. Cells were held at − 60 mV after whole-cell entry and during recording. Due to the flatter morphology of both NSCs and ECs, which restricted the maximum poking distance, we implemented modifications to the poking protocol. Poking probes were made by fire-polishing the tip of glass pipettes to a relatively larger size of ~4 to 6 to achieve sufficient

mechanical stimulation despite the limited poking distance. Indentation stimuli were delivered by displacing poking probes with a piezo-electric actuator (P-841, Physik Instrumente) controlled by Clampex via an amplifier (E-625, Physik Instrumente). A shallow poking angle of ~40° was used to increase the horizontal poking distance and minimize the vertical poking distance. Before each experiment, the probe was slowly moved until visual deformation of the cell. The probe was then retracted ~1 to 2 µm to set the initial probe position as close as possible to the cell surface without physical contact. Initial Poking distance was 3 µm. If poking induced current was not observed the poking distance was increased to 4 µm and then 6 µm until measurable current was observed.

## Total Internal Reflection Fluorescence (TIRF) microscopy imaging

TIRF microscopy was used to image endogenous PIEZO1-tdTomato and PIEZO1-HaloTag channels at 37 °C. PIEZO1-HaloTag cells were incubated in accordance with the HaloTag Ligand Treatment Protocol above, and washed thrice with phenol red-free DMEM/F12 1:1 (Cat. No. 25116001, Invitrogen) and incubated in imaging solution, composed of 148 mM NaCl, 3 mM $CaCl_2$, 1 mM KCl, 2 mM $MgCl_2$, 8 mM Glucose, 10 mM HEPES, pH 7.30, and 316 mOsm/L osmolarity. The hypotonic imaging solution was made by diluting the solution with 20% MQ water. PIEZO1-HaloTag and PIEZO1-tdTomato samples in Fig. 1, Supplementary Fig. 3, Supplementary Movie 1, Supplementary Movie 2, Supplementary Movie 3, Fig. 2A, B were all imaged using an Olympus IX83 microscope fitted with a 4-line cellTIRF illuminator, an environmental control enclosure and stage top incubator (Tokai Hit), programmable motorized stage (ASI), a PLAPO 60x oil immersion objective NA 1.45, and a Hamamatsu Flash 4.0 v2 + scientific CMOS camera (1 pixel is approximately 0.109 µm, no pixel binning). The laser power for the 560 nm laser in these experiments was 0.43 mW at the back pupil of the objective. All other samples were imaged using an Olympus IX83 microscope fitted with a 4-line cellTIRF illuminator, an environmental control enclosure and stage top incubator (Tokai Hit), programmable motorized stage (ASI), a PLAPO 60x oil immersion objective NA 1.50, and a Hamamatsu ORCA-Fusion BT Digital CMOS camera (1 pixel is approximately 0.108 µm, no pixel binning). To image JF646-BAPTA-labeled cells in this system with the 640 nm laser, the laser power at the back pupil of the objective was 3.5 mW for 100 fps videos; 12 mW for 200 fps videos; and 20.5 mW for 500 fps videos. All images were acquired using the open-source software Micro-Manager[72]. Cells were illuminated with either a 560 nm or a 640 nm laser, as appropriate for the fluorophore used, and images were acquired with a Hamamatsu ORCA-Fusion BT Digital CMOS camera. For TIRF experiments, 1 camera unit (c.u.) is equivalent to 0.229 photoelectrons. Microscope and filter cube information can be found in Supplementary Table 4.

## Analysis of PIEZO1-HaloTag puncta diffusion based on TIRF imaging

To assess the diffusion properties of PIEZO1, the single-molecule localization software package ThunderSTORM[48,49] implemented in FIJI[48,49] was used to detect and localize single PIEZO1-HaloTag puncta observed in TIRF recordings. ThunderSTORM was set to use a multi-emitter fitting. PIEZO1-HaloTag trajectories were then generated by connecting puncta localization centroids over time using the image analysis software FLIKA. The nearest punctum within three pixels (centroid-centroid distance) between adjacent frames was linked and assigned to a trajectory. If no puncta were detected within adjacent frames, the next frame was also searched, and if no puncta could be detected within the search radius, the trajectory was terminated. Tracks with a minimum of 4 links were analyzed to calculate signal-to-background, diffusion coefficients, trajectory path length, and mean squared displacement (MSD) values. Single lag displacements (SLD)

were calculated using the distance moved by the punctum between two consecutive frames. In cases where the punctum was briefly undetected (a "gap frame"), punctum positions in the gap frames were interpolated.

### Evaluation of x-y drift in immobilized PIEZO1-HaloTag

To assess the relative contribution of x-y drift in our puncta diffusion data, we labeled PIEZO1-HaloTag ECs with JF646 HTL. Then we fixed the cells to immobilize the PIEZO1-HaloTag JF646 puncta using a 4% v/v paraformaldehyde solution, including 5 mM MgCl₂, 10 mM EGTA, and 40 mg/mL sucrose in PBS (pH = 7.3) for 10 min at room temperature. Samples were washed 3 times with 1x PBS and imaged with TIRF microscopy. Images were acquired previously as described in the section "Total Internal Reflection Fluorescence (TIRF) Imaging." Immobilized puncta data was then processed using the single-molecule localization software package ThunderSTORM implemented in FIJI. Thunderstorm settings were the same as described in "Analysis of PIEZO1-HaloTag Puncta Diffusion based on TIRF imaging." After processing, we ran the drift correction module to determine how much drift was present in the immobilized data. We found a negligible contribution of x-y drift, <0.1 pixels (10 nm) over 5 s.

### Calculation of localization error using fixed PIEZO1-tdTomato and PIEZO1-HaloTag samples

To compare the relative localization error between PIEZO1-tdTomato and PIEZO1-HaloTag samples, we first harvested mouse PIEZO1-tdTomato mLSECs as described in the Methods section "Mouse Liver Sinusoidal Endothelial (mLSEC) Isolation and Culture." PIEZO1-tdTomato mLSECs and PIEZO1-HaloTag ECs were plated on MatTeks. Cells were then fixed using a 4% v/v paraformaldehyde solution, including 5 mM MgCl₂, 10 mM EGTA, and 40 mg/mL sucrose in PBS (pH = 7.3) for 10 min at room temperature. Samples were washed 3 times with 1x PBS and imaged with TIRF microscopy. Images were acquired as described in the section "Total Internal Reflection Fluorescence (TIRF) Imaging." Immobilized trajectories were extracted using the Methods section "Analysis of PIEZO1-HaloTag Puncta Diffusion based on TIRF imaging." Localization error was determined by measuring the distance of each puncta's localization from the mean position of the trajectory across records from fixed samples.

### Comparison of PIEZO1-tdTomato and PIEZO1-HaloTag signals

To compare fluorescence intensities from PIEZO1-tdTomato and PIEZO1-HaloTag JF549 HTL videos, the sum of all pixel intensities in a region of interest (integrated intensity) was measured over the duration of the video in FIJI. This was done for 3 ROIs in each video. ROIs were 71 × 71 pixels (7.7 × 7.7 μm). These values were then background-subtracted using the integrated intensity from an ROI outside the cell border of the same video. ThunderSTORM was used to identify puncta in the first frame of each of the three ROIs, and the background-subtracted integrated intensity was scaled for the number of puncta. These scaled integrated intensity values from each of the 3 ROIs within a video were then averaged to create a video-level average, which were then again averaged over 20 videos for PIEZO1-tdTomato and 19 videos for PIEZO1-HaloTag to yield an overall mean for each fluorophore.

To calculate the relative signal-to-background ratio for JF549-labeled PIEZO1-HaloTag and for PIEZO1-tdTomato puncta (for Fig. 2A), puncta were detected in the first frame for each video. The punctum signal intensity was measured from a 3 × 3 pixel ROI after subtracting the camera black level. Background fluorescence was similarly determined from a 3 × 3 pixel ROI inside the cell devoid of puncta.

### Analysis of PIEZO1-HaloTag activity based on TIRF imaging

FLIKA software was used to build trajectories from PIEZO1-HaloTag cells labeled with JF646-BAPTA and to record puncta fluorescence

intensity within a 3 × 3 pixel ROI for each frame[50]. To account for the flickering behavior of the JF646-BAPTA HTL, the allowable gap time was 90 ms. Intensities and positions were interpolated over gap frames. Puncta intensities were background-subtracted using intensities from a 3 × 3 pixel ROI inside the cell devoid of puncta. After trajectory building, videos were visually inspected to identify representative stationary and mobile puncta.

### Puncta density analysis

To quantify the density of PIEZO1 puncta labeled with JF646-BAPTA or JF646 HTL, we first binned the videos acquired at either 100 fps (Fig. 3 and Supplementary Figs. 1, 2) or 200 fps (Fig. 4) to 10 fps, then cropped a region of interest from the first frame of the video. Using the same settings as described in the Methods section "Analysis of PIEZO1-HaloTag Puncta Diffusion based on TIRF imaging", we detected and localized PIEZO1 puncta. The number of puncta in the cropped frame was recorded and scaled to the area of the cropped region of interest.

### Calcium calibration experiment in transfected WTC-11 hiPSCs

One million WTC-11 iPSC cells were transfected with 3 μg of the plasmid pCDNA5/FRT/TO_HaloTag7_T2A_EGFP (Addgene No.169325) using AMAXA Stem Cell Kit No.1 (Cat. No. VPH-5012) and Nucleofector II with the B-016 program. pCDNA5/FRT/TO_HaloTag7_T2A_EGFP was a gift from Kai Johnsson (Addgene plasmid No. 169325)[73]. 50,000 cells were plated on MatTek plates in mTESR Plus media containing CEPT cocktail (50 nM Chroman 1, 5 μM Emricasan, 0.7 μM trans-ISRIB (Captivate Bio, Cat. No. CET01B), 1x Polyamine Supplement (Sigma-Aldrich, Cat. No. P8483-5ml) and cultured overnight at 37 °C in a humidified incubator with 5% CO₂. GFP-positive cells indicated successful transfection and expression of HaloTag7 under the control of the CMV promoter. Transfected cells were subsequently labeled with a mixture of 0.5 nM JF646-BAPTA and 0.5 nM JF549 HTLs. Then, cells were fixed in a 4% v/v paraformaldehyde solution, including 5 mM MgCl₂, 10 mM EGTA, and 40 mg/mL sucrose in PBS (pH = 7.3) for 10 min at room temperature, permeabilized in 0.3% Triton X-100 in PBS for 5 min, and imaged with sequential changes of bath solution containing 0 nM, 75 nM, and 39 μM Ca²⁺ solutions (Calcium Calibration Kit #1, Invitrogen, Cat. No. C3008MP) at 23 °C. The Calcium Calibration Kit #1 included two 50 mL solutions: 10 mM K₂EGTA and 10 mM CaEGTA. Both solutions were prepared in deionized water and contained 100 mM KCl and 30 mM MOPS, with a pH of 7.2. The two solutions are mixed to give the desired Ca²⁺ concentration. Cells were imaged with TIRF microscopy at 10 fps. Fixed puncta fluorescence intensity within a 3 × 3 pixel ROI was recorded and background subtracted, as previously described in the section "Analysis of PIEZO1-HaloTag activity based on TIRF imaging".

### Live Micropatterned Neural Rosette (MNR) imaging

MNRs of 140 μm diameter grown on 19 mm square Arena A CYTOO chips (Cat. No.10-020-00-18) were imaged on a modified adaptive optical lattice light-sheet microscope (AO-LLSM)[61]. Prior to all imaging, the microscope was calibrated to correct for optical aberrations from the system. The Cytoo chip was mounted on a custom-designed sample holder and immersed in 40 ml of phenol-free MNR culture medium[74]. The excitation (Thorlabs 0.6 NA, TL20X-MPL) and detection (Zeiss 1.0 NA, 421452-9800-000) objectives were also immersed into the imaging medium. The MNRs were imaged with a multiBessel square lattice light sheet with the NAsq of 0.35 and a 0.3/0.4 NA annular mask. The 488 nm, 560 nm and 642 nm lasers were used to visualize SPY505-DNA (DNA, Spirochrome Cat. No. CY-SC101), SPY555-actin (actin, Spirochrome Cat. No. CY-SC202), and Janelia Fluor 635-HaloTag Ligand (Non-Bapta) or Janelia Fluor 646-BAPTA-AM- HaloTag ligand (PIEZO1), respectively, with power at the back pupil of the excitation objective of 45 μW for 488 nm, 50 μW for 560 nm and ranging between 2.1–2.9 mW for 642 nm. To balance volumetric

imaging speed, signal to noise and photobleaching, images were acquired using camera exposure between 30–50 ms. All data from the AO-LLSM were collected on two Hamamatsu ORCA-Fusion sCMOS cameras. Emission light from actin, DNA, and JF635 or JF646 was separated by a dichroic mirror (Chroma T600DCRB) and passed to two cameras equipped with either a Semrock FF01-600/37-25 emission filter for actin or a Semrock FF01-538/685-25 filter for DNA and JF635 or JF646-BAPTA. Microscope and filter cube information can be found in Supplementary Table 4.

The 3D volumetric imaging of MNRs was performed by tiling AO-LLSM across the sample. Each tile (approximately $200 \times 100 \times 15\,\mu m^3$) was scanned by moving the sample stage at 400 nm step sizes. Each neighboring tile had a 5 μm overlap between each adjacent tile. Mismatch between the excitation LLS and the detection focal plane caused by the MNR were corrected prior to every acquisition and was essential to ensure optimal imaging of PIEZO1, DNA, and actin structures[75]. The AO-LLSM data was processed using MATLAB versions R2022b and R2023a. The large 3D MNR volumes were stitched in skewed space, deconvolved, deskewed, and rotated on the Advanced Bioimaging Center's computing cluster at UC Berkeley using the computational pipelines published on GitHub[76]. The skewed space deconvolution was performed[74] with experimentally measured point spread functions obtained from 200 nm fluorescent beads (Invitrogen FluoSpheres Carboxylate-Modified Microspheres, 505/515 nm, F8811[74]. The nuclei were denoised using Content-Aware Image Restoration (CARE)[74,77]. The training data for the denoising model was collected using lattice light-sheet microscopy as previously described[74] by volumetrically scanning LLC-PK1 cells expressing a nuclear marker (H2B) to record low SNR and corresponding high SNR 3D stacks. The AO-LLSM instrument was controlled using a custom LabVIEW based image acquisition software (National Instruments, Woburn, MA) licensed from Janelia Research Campus, HHMI.

To observe the dynamics of PIEZO1-HaloTag in MNRs, 3-plane stack videos were acquired. For this, the sample stage (using the SmarAct MLS-3252 Electromagnetic Direct-Drive) was rapidly scanned across a 1 μm sample range (comprised of three image planes spaced 400 nm along the sample stage scan axis, corresponding to 215 nm along the optical z axis) at an interval between 160–210 ms per stack for 30 time points. The dynamic time series datasets were deskewed and maximum intensity projected prior to analysis[78]. The corresponding DNA and actin volumes were recorded at the same location prior to acquiring the PIEZO1-HaloTag time series. A new dataset of 30 time points was collected every ~10 μm throughout the MNR and was repeated on multiple samples.

### Detection of PIEZO1-HaloTag JF635 puncta and separation from spurious detections

The maximum intensity projection (MIP) images of deskewed 3-plane stacks were used for puncta localization analysis. PIEZO1-HaloTag puncta localization was analyzed in JF635-labeled PIEZO1-HaloTag MNRs. JF635-labeled PIEZO1-HaloTag KO MNRs and unlabeled PIEZO1-HaloTag MNRs served as controls to characterize and subsequently filter out spurious detections due to autofluorescent spots, remains of unbound JF635 probe, or local background fluctuations being detected as puncta, as described below.

The ImageJ plugin ThunderSTORM was used to detect PIEZO1 puncta, generating puncta localization maps for each frame of the acquisition video. A Gaussian function was fitted to the detected puncta spots to generate a centroid for each punctum, as described above for TIRF data. To distinguish labeled PIEZO1-HaloTag puncta from autofluorescent spots, which were much larger in size and also present in PIEZO1-HaloTag KO MNRs and unlabeled PIEZO1-HaloTag MNRs, a first filtering step was applied by setting a threshold based on the standard deviation ($\sigma$) of the Gaussian fit of each punctum. Puncta

with $\sigma$ higher than 280 nm were removed since experimental $\sigma$ measured on fluorescent beads (diameter: 0.2 μm, excitation 642 nm) was ~189 nm.

In order to remove any spurious detections due to unbound JF635 dye passing through the plane of imaging during acquisition, or to ThunderSTORM erroneously detecting small local background fluctuations as puncta, a time-based filtering step was applied to retain only objects that persisted over multiple frames. The custom-built FLIKA algorithm was used to link detected puncta across frames with a maximum linking distance between consecutive frames of 324 nm (3 pixels) and a linking gap of one frame, as done for TIRF data analysis above. Tracks composed of at least 3 segments (i.e., successfully linked puncta in at least 4 frames) were considered representative of bona fide PIEZO1-HaloTag puncta and retained for further analysis. See the Supplementary Results section for the performance of these settings in detecting and filtering PIEZO1-HaloTag puncta.

### Detection of PIEZO1-HaloTag JF646-BAPTA signal and separation from spurious detections

Puncta in MNRs labeled with JF646-BAPTA HTL were detected using ThunderSTORM as described above for JF635-labeled MNRs. The same size-based filtering step as done for the JF635-labeled samples above was applied, which removed spurious detections in autofluorescent spots with a $\sigma$ over 280 nm.

The fluorescence intensity of the JF646-BAPTA HTL depends on the local $Ca^{2+}$ concentration, which changes with the channel's activation state. Thus, the fluorescence intensity of the JF646-BAPTA HTL is higher when PIEZO1 channels are active/open than when in the resting/closed state; as channels flicker between open and closed, the fluorescence intensity of puncta varies over time (e.g., see kymograph in Fig. 5F). Thus, the time-based filtering used for the JF635 HTL above cannot be applied for image stacks of JF646-BAPTA HTL. Hence, control JF 646-BAPTA PIEZO1-HaloTag KO and unlabeled PIEZO1-HaloTag MNRs were used to determine the threshold intensity values for separating puncta associated with active/open channels from unbound HTL and small autofluoresence puncta. Based on the distribution of integrated intensity of puncta in the control samples, ~77 photons (261 c.u.) was selected as the filtering threshold. Puncta in JF646-BAPTA PIEZO1-HaloTag MNRs below this threshold were filtered out, while puncta with intensities above this value were retained for analysis. See the Supplementary Results section for the performance of these settings.

### Determining distance of PIEZO1-HaloTag puncta to MNR lumen and outer edge

In order to determine the distances between detected PIEZO1 puncta and the lumen or outer edge of the MNR, we manually drew masks identifying the lumen border and the outer edge of the MNRs on ImageJ, using the actin labeling as reference (example in Fig. 5D).

For the JF635-labeled PIEZO1-HaloTag experiments, we computed the mean position coordinates of each PIEZO1-HaloTag track. We then computed distances of this mean position for each track to the MNR lumen and outer edge masks using Euclidean distance transform, and rounded the distances up to the nearest pixel coordinates. The puncta overlapping with the masks were assigned a distance of zero for the corresponding mask. For the JF646-BAPTA-labeled PIEZO1-HaloTag signal, we used each filtered punctum localization as described above, and Euclidean distances from these to the lumen and outer edges were similarly calculated.

The calculated distance distributions were visualized using density scatter plots. We used the "ksdensity" function in MATLAB with a "normal" kernel having a bandwidth of [2.503 μm, 3.661 μm] for the distances to the lumen and outer edge, respectively. Once the

underlying distribution of the distances is obtained, the scatter plot was drawn using the "Scatter" object in MATLAB, with the colors corresponding to their probability densities. The transparency of the scatter points in the density scatter plots is based on the density values, where darker regions correspond to higher concentrations of the puncta. This normalization method ensures similar smoothing and colormap scaling of the underlying distributions across all experiments, thus allowing us to compare distribution patterns across all the conditions tested.

The individual relative frequency distributions were plotted using histograms computed from the corresponding distance distributions with a bin size of 2 µm. For the histogram plots, the actual frequency of the detections at a particular distance from the corresponding mask was normalized using the total count, along with the bin size, so that the total area is 1. These plots were generated using the "hist" and "stairs" functions in MATLAB. Also, for further analysis, the respective cumulative distribution frequency (CDF) graphs were plotted using the "ecdf" function in MATLAB on the distance distributions, which computes the empirical CDF values from the data.

### 3D detection and filtering of PIEZO1-HaloTag puncta in volumetric samples

The PIEZO1-HaloTag puncta localization in the volumetric data was inspected by numerically fitting a model of the PSF approximated by a 3D Gaussian function[78]. JF635 PIEZO1-HaloTag KO MNRs were used as a control to determine filtering parameters to exclude autofluorescent blobs present in the data. For filtering, thresholding was applied on the fitted amplitude of the intensity above the local background, as well as the fitted local background, to extract the detected diffraction-limited puncta. This approach ensured that most puncta in the Knockout sample were removed, while retaining most of the puncta in JF635 PIEZO1-HaloTag MNRs.

### Statistics & reproducibility

Sample sizes and number of biological replicates are indicated in the corresponding figures. For all experiments performed in this study, we did not conduct formal sample size calculations. Each experiment was repeated independently at least three times to ensure reproducibility of our findings. This approach aligns with standard practices in the field for similar mechanistic and imaging-based studies, where biological variability and technical reproducibility are primary considerations. Unless otherwise stated, data in graphs are shown as mean ± SEM. No data points were excluded from the analyses, and the investigators were not blinded to allocation during experiments and outcome assessment. For drug treatments or mechanical stimulation experiments, identical dishes were randomly assigned as Control or Treated. An online estimation stats tool (https://www.estimationstats.com)[79] was used to calculate Cohen's *d*. A normality test was performed prior to all statistical tests and subsequent *p*-value calculations. Normally distributed data were analyzed using the Student's two-sample t-test statistical analysis, while non-normally distributed data were analyzed using the Mann-Whitney test. The specific statistical tests used for *p*-value calculations are indicated in the figure legends. All *p*-value calculations and graphs were performed using OriginPro 2020 (OriginLab Corporation).

### Figure preparation

Figures were prepared using Adobe Illustrator. Schematics used in the figures were prepared with open-source vectorized files from Bioicons (https://bioicons.com/?query=objectiv), Adobe stock images, or by constructing them de novo.

### Reporting summary

Further information on research design is available in the Nature Portfolio Reporting Summary linked to this article.

## Data availability

Methods, representative movie files, and supplementary information are included in the manuscript. Minimum datasets and analysis code for the lightsheet imaging experiments can be found on Dryad[80], and source raw data for supplementary TIRF videos can be found on Zenodo[81]. Complete raw datasets, including source images and analyzed trajectories, included in this study are available upon request from the corresponding author. Source data for graphs are provided in this paper.

## Code availability

In preparation of this work, we used the following software packages: OriginPro 2022 (64-bit) SR1 9.9.0.225, FLIKA version 4.8.1, ImageJ 2.9.0/1.53t; Java 1.8.0_322, Micro-Manager 2.0, cellTIRF 1.4, Python 3.11.5, Imaris x64: 10.0.0, Amira 3D 2021.1, MATLAB versions R2022b and R2023a. Custom analysis code with minimum datasets have been deposited on Github[76,82].

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

## Acknowledgements

We thank Dr. Ian Smith, Ms. Elaine Lai, and Ms. Vivian M. Leung for technical support and the members of the lab for comments on the manuscript. We gratefully acknowledge Dr. Luke Lavis, Janelia Research Campus, and the Janelia Materials project team for sharing Janelia Fluor HaloTag Ligands. We thank Dr. Francesco Tombola, University of California, Irvine, for the use of his patch clamp equipment; and Ms. Allia Fawaz and the Core Facilities of the Sue and Bill Gross Stem Cell Research Center (supported in part through a California Institute of Regenerative Medicine Shared Research Lab Grants (SRL): Cl1-00520). We appreciate the support of the UC Irvine CIRM Shared Resources Laboratories to Enhance In Vitro Stem Cell Modeling and Training grant INFR6.2-15368. We thank Dr. Xiongtao Ruan and Matthew Mueller, UC Berkeley, for helpful discussions on data analysis and visualization.

This work was supported by National Institutes of Health grants DP2AT010376 and R01NS109810 to M.M.P.. G. A. B. was supported in part by NIH T32 NS082174 and the University of California, Irvine Stanley Behrens Fellowship in Medicine, I.C and E.L.E. were supported by the California Institute for Regenerative Medicine (CIRM) under Award Number EDUC4-12822, A.T.L. was supported in part by NIH F31 1F31NS127594 and the University of California, Irvine Graduate Dean's Dissertation Fellowship, and J.R.H. by the HHMI Gilliam Diversity Fellowship. G.L., S.S., and S.U. are funded by the Philomathia Foundation. S.U. is funded by the Chan Zuckerberg Initiative Imaging Scientist program. S.U. is a Chan Zuckerberg Biohub – San Francisco Investigator. The content of this publication is solely the responsibility of the authors and does not necessarily represent the official views of CIRM or other funders.

## Author contributions

J.L.N., M.M.P. (Panicker), I.P., and M.M.P. (Pathak) conceived the idea and designed the project. G.A.B., I.C., E.L.E., J.L.N., G.D.D., G.L., S.Y., E.E.H., M.M.P. (Panicker), S.U., I.P., and M.M.P. (Pathak) developed and designed the methodology and created experimental models. G.A.B., I.C., E.L.E., J.L.N., G.D.D., G.L., A.T.L., J.R.H., and T.D.W. performed the experiments and collected data. G.A.B., I.C., E.L.E., G.D.D., S.S., and T.D.W. analyzed the data. J.L.N., J.J.L., M.M.P. (Panicker), S.U., I.P., M.M.P. (Pathak) provided key materials, equipment, and tools essential for the research. G.D.D. and S.S. developed the software for data analysis. G.A.B., I.C., E.L.E., S.U., I.P., and M.M.P. (Pathak) wrote the original manuscript draft. G.A.B., I.C., E.L.E., J.L.N., G.D.D., S.S., A.T.L., S.U., I.P., and M.M.P. (Pathak) reviewed and edited the manuscript. E.E.H., J.J.L., S.U., I.P., and M.M.P. (Pathak) supervised the project. G.A.B., E.L.E., J.L.N., and M.M.P. (Pathak) administered the project. S.U., I.P., and M.M.P. (Pathak) acquired the funding. All authors read and approved the final manuscript.

## Competing interests

The authors declare no competing interests.
