## [Transparent Peer Review file · Nature Communications]

Visualizing PIEZO1 Localization and Activity in hiPSC-Derived Single Cells and Organoids with HaloTag Technology

Corresponding Author: Professor Medha Pathak

Version 0:

Reviewer comments:

Reviewer #1

(Remarks to the Author)

Bertaccini et al. established a system for endogenously expressing the mechanosensory channel PIEZO1 tagged with a Halotag-based reporter to track this essential mechanosensory molecule in human cells and organoids. The authors utilized state-of-the-art imaging systems, including TIRF super-resolution and light-sheet imaging systems combined with the FIJI plugin Thunder-STORM imaging analysis system, enabling imaging of PIEZO1-Halo at the molecular level in both differentiated stem cells and organoids. Additionally, the authors also applied a calcium-sensitive Halotag ligand to capture PIEZO1 activity in vivo.

The study is well-documented, with particularly stunning imaging data. The data collection and analysis were carefully performed with great detail, as well as with proper controls incorporated in each experiment. This novel imaging approach will significantly contribute to PIEZO's physiological and cellular study and can be readily applied to studying other important mechanotransductive or transmembrane channels."

Before recommending the publication of this work in Nature Communications, several questions and concerns should be addressed:

- 1) It would be beneficial to provide citations to support the statement, "In addition, non-human model organisms used in many studies examining physiological roles of PIEZO1 may not fully recapitulate channel behavior in human physiology."
- 2) Clarification is needed regarding the interpretation of the light/dark blue colors represented in Figure 1C. This information needed to be included in the legend.
- 3) Insight into whether the electrophysiological performance of PIEZO-HT or PIEZO-WT was affected by the addition of Halo-tag ligands would be valuable for understanding the functional implications of the tagging approach.
- 4) The size of positive puncta seems quite different in different types of stem cells. Does stem cell differentiation affect puncta formation, particularly when comparing the images in Figure S3A-B? Is the PIEZO1-HALO form aggregated puncta in cells such as neural stem cells? It would be worthwhile to quantify the size of such puncta in different types of cells to determine whether the pattern is cell-autonomous. This raises another question: Is PIEZO1-Halo puncta formation affected when inducing different types of stem cells?
- 5) What is the cellular mechanism for immobile puncta vs. mobile puncta? Is it due to the newly synthesized, labeled PIEZO1-Halo being anchored on the ER to make it immobile? It would be helpful to add other organelles markers, such as ER markers for imaging, which could help clarify such distinct groups of puncta.
- 6) Since mobile puncta move quickly, how will the authors optimize the imaging analysis system to exclude the possibility that another mobile puncta moves into the selected ROI of the designed immobile puncta when analyzing the intensity? Also, when testing the calcium-sensitive ligands, were the tested puncta localized near the ER structure or on the plasma membrane? Did the cellular locations of the PIEZO1-Halo puncta matter when doing the activity analysis?
- 7) The imaging system works quite well for the wild-type PIEZO1 molecule. As a significant claim and potential application of the established system to the clinical/physiological studies, the authors should at least test one gain-of-function or loss-of-function pathogenic PIEZO mutation using the imaging system to define the sensitivity and resolution of the system when assessing the effect of toxic or pathogenic variants on PIEZO channel activity or molecular dynamics.

(Remarks on code availability)

N.A.

Reviewer #2

(Remarks to the Author)

Summary of Key results:

The authors have worked towards creating another approach to visualize PIEZO1 localization and activity through HaloTags in human induced pluripotent stem cells (hiPSC). By editing both alleles of PIEZO1 to be tagged with HaloTag in hiPSC the authors have ensured the ability to investigate all PIEZO1 activity without overexpressing the protein. This in combination with a suite of HaloTag ligands allows for a variety of unique analyses of PIEZO1 activity in a variety of cells or tissue in vitro. Using these newly developed tools, the authors have been able to validate the ability to visualize PIEZO1 in cultured cells. Additionally, the authors have found both mobile PIEZO1 puncta and immobile PIEZO1 puncta. The authors claim that these tools allow for new approaches to understanding PIEZO1 function in a physiologically relevant context.

Major Comments:

1. The creation of an endogenously labeled PIEZO1 with HaloTag in hiPSCs is a great toolset that opens up the ability to study PIEZO1 activity and localization in a variety of new ways. This is particularly exciting in the context of human disease modeling and large-format drug screenings in cell culture or organoids.

2. There are a few areas where the authors claims are a bit too strong for what they are accomplishing. In particular, the two following examples:

On page 2,

"...In addition, non-human model organisms used in many studies examining physiological roles of PIEZO1 may not fully recapitulate channel behavior in human physiology. Here we present a novel platform to overcome these challenges and to advance physiologically and clinically relevant research on human PIEZO1."

While parts of this statement are true (non-human model organisms do not fully recapitulate channel behavior in human physiology), I would argue that studying physiological roles of PIEZO1 in vitro also does not fully recapitulate channel behavior in human physiology. The in vitro environment will not fully recapitulate all the mechanical stimuli that cells would experience in the human body. The tools described in this work partially overcome the challenges outlined and would benefit from being used in-parallel with non-human model systems.

On page 11,

"...our investigation examined its localization and activity during early neural development." Technically the authors did not study this during early neural development. This was done in a neural organoid model in an attempt to mimic early neural development. This statement could be slightly altered to not overstate the work that was accomplished. A good example of this is in the following sentence where the authors write, "...we employed an hiPSC-derived neural organoid model, MNRs, representing the developing neural tube."

3. While the authors have been able to validate the newly created tools and show some utility, this work struggles to significantly advance our knowledge of PIEZO1 activity and localization in cell or tissue physiology. It seems that the novel findings from this tool is the ability to identify PIEZO1 puncta that are mobile (something that the authors cite as already being known from their own previous work) and the distribution/activity of PIEZO1-HaloTag in MNRs. While these findings are interesting, there is much left to be understood of their importance. What is the significance of mobile PIEZO1 puncta? Is PIEZO1 activity in MNRs modulated by actomyosin forces? This work would be greatly enhanced by showing the importance of the highlighted results. By answering these questions, the authors will be able to show that this newly developed tool is able to produce significant findings that advance our understanding of PIEZO1 in cell or tissue physiology.

4. Page 4, Figure 2B – "the faster imaging afforded by the PIEZO1-HaloTag approach opens up avenues for mechanistic studies on PIEZO1 localization and dynamics during cell migration and other physiological processes". This is another example where the authors could take the time and show what new dynamics and mechanisms can be studied with this tool that previously couldn't be studied. The authors should be able to do this analysis considering they have the tools to investigate this process with PIEZO1-HaloTag and compare to the limitations of PIEZO1-tdTomato.

5. Throughout the manuscript, the authors note that this new PIEZO1-HaloTag technology is a significant improvement over previously existing approaches using PIEZO1-tdTomato. This claim would be greatly improved if the authors included direct comparisons to PIEZO1-tdTomato that showed statistically significant improvements to studying PIEZO1. There are a few sections throughout the manuscript where additional statistical comparisons would be helpful:

a. In Figure 2A, is this increased photostability in the Piezo1-HaloTag puncta significantly greater than PIEZO1-tdTomato? The differences shown in the graph were noted, but there is no statistical test comparing the two groups.

b. In Figure 2B, the authors show PIEZO1-HaloTag in a migrating neural stem cell. In the text, this is said to show improved image quality over PIEZO1-tdTomato, but this is not explicitly shown in the manuscript. Nor is there any statistical analysis to support significant improvement.

c. In Figure 3E & 3H, there is a noted increase in the occupancy in active state of the Yoda1 treated group compared to the DMSO treated group, but this finding could be further strengthened with a statistical comparison to show the significance of this result.

Minor Comments:

1. Figure 1F – Including labeling in the figure that these images are showing PIEZO1-HaloTag labeled with JF646 would

benefit the reader.

2. Figure 1D – The authors can statistically show that P1 KO significantly reduces electrical current following mechanical stimulation. Currently the authors do not show a statistical comparison of P1 WT and P1-HT. Including the statistical results for the comparison between P1 WT and P1-HT would be nice to show that P1-HT isn't significantly altering the channel activity.
3. Figure 2A – Including a label of what the red and grey lines represent in the graph would benefit the reader.
4. Referencing the supplemental videos in the text would be another benefit to readers. Currently supplemental videos are mostly cited in figure legends. For example, referencing supplemental video 4 on page 6 would give the readers an example of mobile puncta.
5. Figure 4G – Rather than writing "Same as E but..." in the figure legend, explicitly stating the information would be helpful to readers. Additionally labeling on the graph to note that this is with JF646-BAPTA HTL would also be helpful for readers.

(Remarks on code availability)

Reviewer #3

(Remarks to the Author)

Bertaccini et al.'s manuscript is an interesting integration of the HaloTag method for visualising, tracking PIEZO1 in cultured cells as well as neuronal organoids, and for tracking local calcium. It is a nice demonstration of the principle of this approach, however pivots heavily on a couple of assumptions that need validating. I have proposed a number of major points that I think need to be addressed before I would recommend this for publication.

Major:

1. Page 4, paragraph 2: The JF HaloTag benchmarking against the tdTomato is fairly convincing. However, JF dyes are primarily designed for photoswitching experiments, hence the photobleaching can be reversible. For them to be used in experiments showing recovery of renewal of PIEZO1, authors will need to demonstrate the extent of the non-recovery from the dark state (after a dark/non-excited period). Until then, it is impossible to discern this fluorescence recovery (e.g. Figure 2b) from the recovery of JF646 from their triplet/dark states.
2. Page 6, paragraph 2: I am surprised that even after fixation, the authors' tracking approach detects a mean path length peaking around 1.2 micron - particularly as the localisation uncertainty is 20 nm. I suspect the 3-pixel tracking radius criterion is too broad. I would like to see this tracking protocol benchmarked and optimised against image sequences completely immobilised (in vitro, and drift corrected) single dye ligands. Until then, I am not convinced about the (particularly large) track lengths and diffusion co-efficients derived for PIEZO1 here.
3. The data presented in Figure 2, in my view, are an over-simplification of the PIEZO1 tracking outcomes.
 - a. Firstly, the authors use linear fits to the first couple of seconds of the MSD/t curves to obtain the diffusion coefficients. However those curves are clearly non-linear, which to me suggests that a straightforward passive diffusion process is not likely to explain the PIEZO1 mobility. It is possible that there is a cumulative effect of drift in the sample which does not seem to have been addressed explicitly. Assuming that drift was negligible, the non-linear nature of these curves indicates a subdiffusive movement that has neither been explored nor ruled out in the paper.
 - b. Secondly, the multimodal Gaussian curve fit approach to the path lengths histogram is not particularly convincing based on the data provided here. Even for a passive Levy walk, typical track lengths of 4-6 microns over just 5 seconds for a macromolecular complex such as PIEZO1 is fairly high. The authors provide no reference data to PIEZO1 diffusion in 'clean' environments such as reconstituted bilayers. Given the lack of other validating information such as drift correction and event-specific localisation error, I find this data inconclusive.
4. Page 8, paragraph 1: The calcium time traces in the experiment using JF646-BAPTA resemble single channel opening events, however photo-electron calibrated intensity units would be more helpful in discerning whether these are photon-limited events representing single-PIEZO1 openings or indeed small clusters. As a minimum, data like these should be accompanied by an in-vitro calibration of the single molecule (single JF646_BAPTA HaloTags) fluorescence in response to controlled calcium levels which will indicate whether the single molecule-level of calcium sensing is achievable in the imaging conditions used here. Given that JF646 is inherently photoswitchable, the authors have not additionally shown any evidence to eliminate the possibility (or a component) of blinking or radial fluctuations in these fluorescence traces. The supplementary movies included are so saturated that it is impossible to tell how such stepwise fluorescence traces are obtained.
5. Page 19, paragraph 3 "In order to remove any spurious detections due to unbound JF635 dye passing through the plane of imaging...": Is there a specific reason why the excess JF635 Halo ligands were not washed out during imaging? Whilst it has been shown by the Lavis group that the fluorescence is up to 80-fold dimmer when the ligand is unbound, this would still make a non-zero contribution to the background. Whilst the authors go through an elaborate time-based filtering of background fluctuations, I would like to see how much of these spurious detections have been due to the practice of not washing out the excess probe. How much of the background observed in the PIEZO1-HaloTag KO samples was caused by this approach?

Minor:

1. Page 2, The orthogonal assays of positive expression of PIEZO1-HaloTag and PIEZO1 HaloTag KO are compelling, however the protocols of cell lysis and immunoblot have not been detailed. For full transparency, this information needs to

be included either in the main text or supplement.

2. Page 4, paragraph 2: When reporting the protocols for photobleaching experiments, it is imperative that you report the excitation laser power as a function (e.g. in W/cm²). This becomes more crucial when experiments showing so-called single-channel activity are shown with the same imaging probes.

3. Final paragraph of the Results: Authors use 77 photons as the threshold point (based on negative controls). However, there is no information provided on the sample-to-sample (and depth dependent) variation in the background. I find this 'computational' thresholding protocol no more arbitrary than manual eyeballing. This needs to be discussed as a key limitation of the approach.

4. Paragraph 3, page 17: Multiple typographic errors 'rois', need to be revised to 'ROIs'.

(Remarks on code availability)

Version 1:

Reviewer comments:

Reviewer #1

(Remarks to the Author)

Great work! The authors have addressed all my comments and questions. I don't have any further comments.

(Remarks on code availability)

Reviewer #2

(Remarks to the Author)

In this revised manuscript, Bertaccini et. al. have added depth to their study showcasing the utility of their newly developed PIEZO1-HaloTag system as a way of studying endogenous Piezo1 activity. The additional statistical tests, supplementary figures, and additions in figure 4 address many of my concerns from the original submission. I only have a few comments that remain.

Line 127, "...mechanical stimulus imparted by with poking using a blunt glass probe (poking assay)." This phrasing is a little awkward. The awkwardness might be resolved by removing "by".

Figure 3 D, F, and I include expanded tracings in red to the right of the original graph. The time scale on these goes from 0 seconds to 3 seconds (D and F) or 0.5 seconds (I). Because these expanded tracings correspond to the red marked regions on the left graph, should the time scale not also reflect the corresponding time? The authors seemingly did this in Figure 4E, although the timescale on the expanded tracing only fits the neural stem cell graph.

Aside from differences in PIEZO1 expression levels, do the authors have any additional insights as to why NSC's have fewer and less active puncta than ECs? These results showcase the utility of your new tool, but is there additional significance or importance to these findings? Showing the significance or importance of this data will in turn show the impact of this study and tool on the field.

In Figure 4A, the area surrounding the NSC has several puncta that are a similar size and brightness to those noted inside of the cell. Are these puncta a result of high background intensity or are these puncta from surrounding cells?

Lines 327-341, Using PIEZO1-Halotag to investigate PIEZO1 response to osmotic stress is an innovative approach that showcases more utility for this developed tool. The methodology for changing osmolarity is currently hard to find in the Methods section. Under the Total Internal Reflection Fluorescence (TIRF) microscopy Imaging subheading there is information on the solution with osmolarity of 316 mOsm/L. I currently do not see information on the 251 mOsm/L hypotonic solution in the methods section.

(Remarks on code availability)

Reviewer #4

(Remarks to the Author)

Bertaccini et al. utilized human induced pluripotent stem cells (hiPSCs), CRISPR/Cas9 genome editing, and the well-established HaloTag technology to develop advanced imaging and analysis workflows for investigating the mechanosensitive PIEZO1 channel. Their study focused on PIEZO1 localization, diffusion dynamics, and channel gating using calcium sensors. This approach was demonstrated not only in different differentiated single cell types but also in a living neural organoid model using advanced light-sheet imaging. They successfully imaged individual PIEZO1 channels (punctae), analyzed their mobility and diffusion through single-particle tracking, and demonstrated the superiority of their

method over previous studies based on the fluorescent protein tdTomato. Notably, the application of Janelia Fluor JF646-BAPTA-3'AM probes for high-frame-rate calcium imaging represents a significant contribution to the manuscript.

The authors have made significant improvements to the first version of the manuscript, addressing and clarifying most of the original concerns raised by the former reviewer #3. I am in favor of publication but would like to suggest a few additional minor modifications. Below are my point-by-point comments on the authors' responses to the former reviewer:

(1) I agree that this was likely a misunderstanding by the reviewer and the mentioned references concerning the JF dyes are correct.

(2) Using the total path length over a defined time period (here, 5 seconds) as a criterion for identifying immobile particles, while valid, may have led to a misunderstanding by the previous reviewer. It could also be challenging for readers without a deeper understanding of single-particle tracking to interpret. The total path length of an immobile particle increases monotonically at each step due to localization errors. Moreover, presenting the cumulative histogram of squared displacements (Fig. 2F) does not effectively highlight the presence of immobile particles. Sample drift was correctly addressed by the authors and appears to be negligible.

First, I suggest including a movie of the raw data with overlaid trajectories within a smaller ROI, such as a portion of the cell shown in Supplementary Movie 4 (Fig. 2C), to demonstrate that the tracking works effectively. Additionally, I recommend presenting a histogram of single-step length distributions (neither squared nor cumulative) and highlighting the region corresponding to localization precision, which should show a distinct peak representing immobile trajectories. Moreover, MSD analysis based on single trajectories, as shown in Fig. 2I, is widely used in the field to differentiate between mobile and immobile particles. I suggest retaining Figs. 2F and 2I while adding the single-step length distribution plot. Figs. 2G and 2I could be moved to the supplementary information to keep the criterion for immobile punctae. Finally, the authors should discuss the fraction of nonspecific binding of HaloTag ligands to the cell surface, which could contribute to immobile signals. Referring to Supplementary Movies 2-4, there appears to be a significant fraction of signals on the glass surface outside the cell area.

(3) a) Using linear fits over the first few seconds of the MSD is common in the field for determining apparent diffusion coefficients. The results are in agreement with the diffusion coefficients of protein complexes in the plasma membrane of living cells.

b) I believe this likely stems from a misunderstanding, where the total track length is confused with the end-to-end trajectory distance, which is significantly shorter in the case of a simple random walk. The authors have appropriately added references to support their findings regarding the measured diffusion coefficients

(4) Regarding this point raised by the previous reviewer, I have a general question about the data interpretation. In the Methods section, the authors stated that all samples were labeled with only 500 pM at 37°C. However, the incubation time, which should be provided in a table (cp. line 648) is missing. At such low concentrations and with a reasonable incubation time (e.g., 30 minutes), one would expect that only a small fraction ($\leq 10\%$) of HaloTags would be labeled (cp. T. Appelhans et al., *Nano Lett.* 2012, 12, 610–616). Although the PIEZO1 channel is a trimer, most channels are likely labeled with only a single dye under these conditions. This should be discussed or clarified and could potentially be determined by analyzing single-puncta bleaching events (as e.g. Hummert J et al., *Mol Biol Cell.* 2021 Nov 1;32(21):ar35. doi: 10.1091/mbc.E20-09-0568 or Danial et al., *The journal of physical chemistry letters*, 13(3), 822–829. <https://doi.org/10.1021/acs.jpcclett.1c03835>) in time-lapse recordings, which should already be available and could be reviewed. What was the rationale behind the low labeling concentration?

I have another general question regarding the Ca²⁺ measurements. The authors state that the BAPTA probe can be used to analyze single-channel gating through direct analysis of intensity fluctuations (lines 262–264 and 297–300). However, the rate of Ca²⁺ chelation and de-chelation should also be considered as a potential rate-limiting step. In particular, the off-rate (de-chelation) could be significantly slower than the closing of the channel. Additionally, examining the intensity traces in Fig. 3D, Fig. 3F, and Supplementary Figures 6–14, a single intensity level - and occasionally two levels - are visible most of the time. This suggests that most channels are labeled with only a single dye. The labeling efficiency and its implications for data interpretation should be more explicitly addressed in the Results and Discussion sections.

(5) Data and Code Availability: Two GitHub repositories are listed and accessible. However, the first repository (<https://github.com/Pathak-Lab/PIEZO1-LocalizationTools>) lacks a minimal example with raw data to test the code, and no hardware requirements or installation instructions are provided. The second repository (https://github.com/abcucberkeley/piezo1_analysis_pipeline) includes a minimal example, but hardware requirements are also missing. It appears that a high-performance workstation is needed, as I was unable to run the code on a smaller workstation."

Minor concerns:

Minor concerns (1) and (3–4) raised by the former reviewer were adequately addressed by the authors. I have one suggestion for point (3):

The authors reported the laser power in units of [W/cm²²²] at the back pupil plane of the objective. However, these units represent power density, not power, and should instead be estimated for the focal plane of the objective. Typical values range from 10 to 300 W/cm², depending on the exposure time (1–30 ms). Alternatively, the authors could report the power in [W] at the back pupil plane, as they do in the AO-LLSM section.

Here are some additional minor points that the authors should address:

(1) Fig. 1B: Please add labels and a legend to the structure, along with a color code.

(2) Terminology: Throughout the text, the authors use both "mobile vs. immobile" and "motile vs. immotile." I recommend using consistent terminology and sticking to one set of terms.

- (3) MSD Analysis in Fig. 2I: I suggest using the term "lag time" for the axis label in the figure and in the text to distinguish lag time from the simple time points of the acquisitions.
- (4) Figure 3: This figure is supported by three movies showing very small ROIs with flickering of the JF646-BAPTA HTL probe. I would like to see a movie with a short sequence of the full cell at a high frame rate (200 fps) to illustrate the flickering of the BAPTA probe in single-cell level.
- (5) Methods Section (line 648): A Supplementary Methods Table No. 1 is mentioned, which should list the incubation times for all JF-HTL dyes, but this information is missing.
- (6) Suppl. Fig. 1: Please add a legend for abbreviations in the figure caption.
- (7) Suppl. Fig. 6: The figure is incomplete/clipped and needs to be corrected.
- (8) Suppl. Information File (line 243): The authors give the threshold in photons here. I suggest sticking to camera units (c.u.) throughout the manuscript or converting all camera units to photons using the given conversion factor.
- (9) Suppl. Figs. 19 & 20: The detections (green circles) in panels B, C, and D are very small and should be adjusted for better visibility.

(Remarks on code availability)

Two GitHub repositories are listed in the manuscript and accessible. However, the first repository (<https://github.com/Pathak-Lab/PIEZO1-LocalizationTools>) lacks a minimal example with raw data to test the code, and no hardware requirements or installation instructions are provided. The second repository (https://github.com/abcucberkeley/piezo1_analysis_pipeline) includes a minimal example, but hardware requirements are also missing. It appears that a high-performance workstation is needed, as I was unable to run the code on a small workstation with Windows 10 and only 16 GB RAM using Matlab R2023b.

Version 2:

Reviewer comments:

Reviewer #4

(Remarks to the Author)

Bertaccini et al. have addressed most of my comments and concerns in the revised manuscript and I support its publication in Nature Communications. I have only three minor comments/suggestions:

- (1) Regarding the degree of labeling using JF-549, JF-646, and JF-635 HaloTag ligands: In the revised manuscript, the authors describe labeling with overnight incubation at 500 pM. In contrast, the JF-646-BAPTA-3AM probe was incubated for only 15 minutes. The authors state that they performed extensive optimization to achieve full labeling with low background of all three HaloTags of the PIEZO1-trimer. Standard live-cell HaloTag labeling protocols typically use 50–100 nM for 15–30 minutes at 37°C to achieve a high degree of labeling. Interestingly, the authors did not follow this approach and instead opted for much longer incubation times. Extended incubations at concentrations above 1 nM generally increase nonspecific binding. The authors should briefly explain their labeling strategy in the methods section and clarify why they did not use standard protocols. Additionally, they should mention in the text that the fluorogenicity of JF-646 (21-fold, Grimm et al., Nat Methods 12, 244–250 (2015)) and JF-635 (113-fold, Grimm et al., Nat Methods 14(10): 987–994 (2017)) was an important factor in reducing fluorescent background during long incubation times or when labeling complex samples such as MNRs, where deep tissue penetration requires extended incubation.
- (2) Some supplementary movies exhibit strong compression artifacts. These should be improved.
- (3) The authors stated that they replaced the terms motile/immotile with mobile/immobile throughout the manuscript. However, both versions still appear in multiple instances. This should be corrected for consistency.

(Remarks on code availability)

Both GitHub repositories "PIEZO1-LocalizationTools" (<https://github.com/Pathak-Lab/PIEZO1-LocalizationTools>) and "piezo1_analysis_pipeline" (https://github.com/abcucberkeley/piezo1_analysis_pipeline) are accessible, provide a minimal example, hardware requirements and installation instructions. I could run the minimal example of the piezo1_analysis_pipeline using Matlab. The pipeline seems to reproduce the data shown in Figure 5. The second toolbox PIEZO1-LocalizationTools provides enough information in the installation instructions and comprehensive guidelines for data processing. However, I was unable to install the FLIKA software, likely due to Python-related compatibility issues with my specific operating system and Python configuration.

We appreciate the reviewers' thorough assessment and thoughtful feedback on our manuscript. Below, we provide a point-by-point response to each suggestion (with the reviewer comment quoted in blue font and our response in black font), outlining how we have revised the manuscript to address the reviewer's concerns. These revisions strengthen our conclusions while maintaining the integrity and scope of our study, which focuses on the validation and application of the PIEZO1-HaloTag hiPSC system. We believe these changes have improved the manuscript's impact, making it a valuable resource for researchers exploring PIEZO1 mechanotransduction in human cells.

Response to Reviewer #1:

Bertaccini et al. established a system for endogenously expressing the mechanosensory channel PIEZO1 tagged with a Halotag-based reporter to track this essential mechanosensory molecule in human cells and organoids. The authors utilized state-of-the-art imaging systems, including TIRF super-resolution and light-sheet imaging systems combined with the FIJI plugin Thunder-STORM imaging analysis system, enabling imaging of PIEZO1-Halo at the molecular level in both differentiated stem cells and organoids. Additionally, the authors also applied a calcium-sensitive Halotag ligand to capture PIEZO1 activity *in vivo*.

The study is well-documented, with particularly stunning imaging data. The data collection and analysis were carefully performed with great detail, as well as with proper controls incorporated in each experiment. This novel imaging approach will significantly contribute to PIEZO's physiological and cellular study and can be readily applied to studying other important mechanotransductive or transmembrane channels."

Before recommending the publication of this work in Nature Communications, several questions and concerns should be addressed:

1. It would be beneficial to provide citations to support the statement, "In addition, non-human model organisms used in many studies examining physiological roles of PIEZO1 may not fully recapitulate channel behavior in human physiology."

We thank the reviewer for their comment. We have now revised this statement and provided references: "Additionally, non-human model organisms commonly used to explore the physiological roles of PIEZO1 may not fully recapitulate the channel's behavior in human physiology (Zheng et al. 2023; Baxter et al. 2020). Here we present a novel platform to overcome these challenges and to provide a versatile human-specific system to complement animal studies, thus advancing physiologically- and clinically-relevant research on human PIEZO1." (lines 60-61).

2. Clarification is needed regarding the interpretation of the light/dark blue colors represented in Figure 1C. This information needed to be included in the legend.

We thank the reviewer for the helpful suggestion to clarify the color coding in Fig. 1C. We have now updated the legend associated with Figure 1C: "Representative traces of cell-attached patch clamp measurements with mechanical stimulation imparted through negative suction pulses for endothelial cells derived from WTC-11, PIEZO1 KO, and PIEZO1-HaloTag hiPSCs. Blue color gradient indicates strength of negative pressure steps associated with suction pulses (light blue: lowest pressure, darkest blue: highest pressure)." (page 3, caption for Fig. 1C)

3. Insight into whether the electrophysiological performance of PIEZO-HT or PIEZO-WT was affected by the addition of Halo-tag ligands would be valuable for understanding the functional implications of the tagging approach.

We thank the reviewer for raising this valuable point regarding functional implications of attaching a HaloTag ligand to PIEZO1. In response, we performed whole-cell patch clamp together

with mechanical stimulation by the poking assay on PIEZO1-HaloTag, PIEZO1-HaloTag + JF646 HaloTag ligand, and PIEZO1-HaloTag knockout. We do not see a difference in the current amplitudes, or inactivation or deactivation kinetics of the channel with attachment of the HaloTag ligand (Supplemental Fig. 4), indicating that attachment of the HaloTag ligand does not disrupt channel function. We have added text describing this data (lines 125-131).

4. The size of positive puncta seems quite different in different types of stem cells. Does stem cell differentiation affect puncta formation, particularly when comparing the images in Figure S3A-B? Is the PIEZO1-HALO form aggregated puncta in cells such as neural stem cells? It would be worthwhile to quantify the size of such puncta in different types of cells to determine whether the pattern is cell-autonomous. This raises another question: Is PIEZO1-Halo puncta formation affected when inducing different types of stem cells?

As the reviewer astutely points out, our data suggests that there may be a difference in the puncta densities and sizes in different cell types. We agree that this is an important question. However, quantifying puncta sizes in our TIRF images would not adequately address this question since puncta span the diffraction limit. Quantitative super-resolution microscopy is needed (e.g. STED and/or MINIFLUX imaging) for accurate quantification of puncta sizes, which is beyond the current scope of this study focused on developing and validating the PIEZO1-HaloTag system. We are actively working on this and will address it comprehensively in a follow-up manuscript.

5. What is the cellular mechanism for immobile puncta vs. mobile puncta? Is it due to the newly synthesized, labeled PIEZO-1-Halo being anchored on the ER to make it immobile? It would be helpful to add other organelles markers, such as ER markers for imaging, which could help clarify such distinct groups of puncta.

The TIRF field will capture puncta in the plasma membrane as well as proximal ER within the evanescent field. In order to determine if distinct behavior of puncta correlates with ER localization, we labeled PIEZO1-HaloTag endothelial cells with JF646 HTL and ER-Tracker Green. We then extracted PIEZO1 trajectories and overlaid trajectories on the ER image (Supp Fig. 5, and main text lines 196-198). Clearly, both mobile and immobile trajectories overlap with both ER and ER-free regions. Therefore, immobile puncta are not solely anchored to the ER and they can also occur in ER free regions. This result aligns with our expectations, as we are using a cytosolic tag that will label all channels in the cell (rather than the surface channels only). We agree that understanding the cellular mechanisms for immobile and mobile puncta is an interesting and important question, and we plan to investigate this in greater detail in a follow-up study.

Supplemental Figure 5. Motility of PIEZO puncta is not determined by association with the ER. Panels show representative TIRF images of multiple positions within PIEZO1-HaloTag

endothelial cells labeled with ER tracker 488 dye (grayscale) overlaid with trajectories of JF646 HTL PIEZO1-HaloTag puncta. Trajectories are color-coded by path length as in Fig. 2 (magenta: mobile, yellow: immobile). Some mobile and immobile trajectories overlap with the ER signal while others localize to ER-free regions.

6. Since mobile puncta move quickly, how will the authors optimize the imaging analysis system to exclude the possibility that another mobile puncta moves into the selected ROI of the designed immobile puncta when analyzing the intensity?

For our initial proof of principle study, we focused on primarily analyzing the activity of immobile puncta that are not interacting with other puncta. This was validated by manual visual inspection of all puncta analyzed. Future work will also examine the activity of mobile puncta in depth.

- a. Also, when testing the calcium-sensitive ligands, were the tested puncta localized near the ER structure or on the plasma membrane?

In this study, we analyzed immobile puncta with the Ca^{2+} -sensitive HTL. As discussed above, using the JF646 HTL, we observed immobile puncta occurring in both ER and ER free regions, suggesting active channels could be present in both cellular compartments (See supplemental Figure 5).

- b. Did the cellular locations of the PIEZO-Halo puncta matter when doing the activity analysis?

This initial study focused on validating this approach to report and quantify PIEZO1 channel activity, and we plan to study the relationship between cellular location and activity in different cell types further in a follow up manuscript.

7. The imaging system works quite well for the wild-type PIEZO1 molecule. As a significant claim and potential application of the established system to the clinical/physiological studies, the authors should at least test one gain-of-function or loss-of-function pathogenic PIEZO mutation using the imaging system to define the sensitivity and resolution of the system when assessing the effect of toxic or pathogenic variants on PIEZO channel activity or molecular dynamics.

[Redacted]

[Redacted]

To illustrate the utility and applicability of our system, we have instead shown how it can be used to systematically compare the expression and activity of PIEZO1 in two different cell types (Fig. 4 and lines 307-326) and also demonstrated that PIEZO1-HaloTag imaging can be used to study the channel's response to an externally-applied mechanical stimulus (Supp. Fig. 12 and lines 327-341).

Response to Reviewer #2:

Summary of Key results:

The authors have worked towards creating another approach to visualize PIEZO1 localization and activity through HaloTags in human induced pluripotent stem cells (hiPSC). By editing both alleles of PIEZO1 to be tagged with HaloTag in hiPSC the authors have ensured the ability to investigate all PIEZO1 activity without overexpressing the protein. This in combination with a suite of HaloTag ligands allows for a variety of unique analyses of PIEZO1 activity in a variety of cells or tissue in vitro. Using these newly developed tools, the authors have been able to validate the ability to visualize PIEZO1 in cultured cells. Additionally, the authors have found both mobile PIEZO1 puncta and immobile PIEZO1 puncta. The authors claim that these tools allow for new approaches to understanding PIEZO1 function in a physiologically relevant context.

Major Comments:

1. The creation of an endogenously labeled PIEZO1 with HaloTag in hiPSCs is a great toolset that opens up the ability to study PIEZO1 activity and localization in a variety of new ways. This is particularly exciting in the context of human disease modeling and large-format drug screenings in cell culture or organoids.

We thank the reviewer for their kind remarks and helpful comments.

2. There are a few areas where the authors claims are a bit too strong for what they are accomplishing. In particular, the two following examples:

On page 2, "...In addition, non-human model organisms used in many studies examining physiological roles of PIEZO1 may not fully recapitulate channel behavior in human physiology. Here we present a novel platform to overcome these challenges and to advance physiologically and clinically relevant research on human PIEZO1."

- a. While parts of this statement are true (non-human model organisms do not fully recapitulate channel behavior in human physiology), I would argue that studying physiological roles of PIEZO1 in vitro also does not fully recapitulate channel behavior in human physiology. The in vitro environment will not fully recapitulate all the mechanical stimuli that cells would experience in the human body. The tools described in this work partially overcome the challenges outlined and would benefit from being used in-parallel with non-human model systems.

We thank the reviewer for their comment. We have edited this statement in the manuscript to "In addition, non-human model organisms used in many studies examining physiological roles of PIEZO1 may not fully recapitulate channel behavior in human physiology (Zheng et al. 2023; Baxter et al. 2020). Here we present a novel platform to overcome these challenges and to provide a versatile human-specific system to complement animal studies, thus advancing physiologically- and clinically-relevant research on human PIEZO1." (lines 59-61).

3. On page 11, "...our investigation examined its localization and activity during early neural development." Technically the authors did not study this during early neural development. This was done in a neural organoid model in an attempt to mimic early neural development. This statement could be slightly altered to not overstate the work that was accomplished. A good example of this is in the following sentence where the authors write, "...we employed an hiPSC-derived neural organoid model, MNRs, representing the developing neural tube.

We thank the reviewer for their suggestion and have revised the sentence on lines 479-481 to “Building upon prior research in a mouse model that reveals PIEZO1's function in neural development¹², we examine the channel’s localization and activity in micropatterned neural rosettes (MNRs), an in vitro model of early human neural tube development.”

4. While the authors have been able to validate the newly created tools and show some utility, this work struggles to significantly advance our knowledge of PIEZO1 activity and localization in cell or tissue physiology. It seems that the novel findings from this tool is the ability to identify PIEZO1 puncta that are mobile (something that the authors cite as already being known from their own previous work) and the distribution/activity of PIEZO1-HaloTag in MNRs. While these findings are interesting, there is much left to be understood of their importance. What is the significance of mobile PIEZO1 puncta? Is PIEZO1 activity in MNRs modulated by actomyosin forces? This work would be greatly enhanced by showing the importance of the highlighted results. By answering these questions, the authors will be able to show that this newly developed tool is able to produce significant findings that advance our understanding of PIEZO1 in cell or tissue physiology.
5. Page 4, Figure 2B – “the faster imaging afforded by the PIEZO1-HaloTag approach opens up avenues for mechanistic studies on PIEZO1 localization and dynamics during cell migration and other physiological processes”. This is another example where the authors could take the time and show what new dynamics and mechanisms can be studied with this tool that previously couldn’t be studied. The authors should be able to do this analysis considering they have the tools to investigate this process with PIEZO1-HaloTag and compare to the limitations of PIEZO1-tdTomato.

Response to pts. 4&5:

We thank the reviewer for their insightful comments and appreciate the opportunity to clarify the scope and focus of our manuscript. The primary aim of this manuscript is to validate and provide proof-of-principle tests of our novel platform, PIEZO1-HaloTag hiPSCs. In the revised manuscript, we have added new data to demonstrate the utility of our approach in studying PIEZO1 activity and localization in cellular physiology.

First, we investigated PIEZO1 activity in two different cell types – endothelial cells and neural stem cells – comparing their relative activity levels. Our results indicate that reduced PIEZO1 activity in neural stem cells compared to endothelial cells arises not only from reduction in its expression level but also from diminished channel activity (Fig. 4, main text lines 307-326). This highlights our tool’s capacity to investigate PIEZO1 across different cellular contexts, offering new insights into the regulation of PIEZO1.

Second, we have broadened the applicability of the PIEZO1-HaloTag system by demonstrating its ability to study the channel’s response to external mechanical stimulation (Supplementary Figure 12, main text lines 227-341). This extends the potential of our tool beyond studying PIEZO1 in innate cellular mechanics, allowing for dynamic mechanistic studies in response to mechanical stimulation.

In addition to these new results, we have performed several controls and added more technical validation of our approach (Sup. Figs. 4, 5 and 6, Supp Fig 19E and 20E,F; main text lines 135-131, 196-198 and 231-264; Supplemental Information lines 174-253). These include a direct comparison of the PIEZO1-HaloTag and PIEZO1-tdTomato systems, which demonstrates the increased precision in studying PIEZO1 localization using the HaloTag approach (Fig. 2A and 2B, lines 148-154).

We agree with the reviewer that the current work raises several important questions, for example, the mechanism and possible functional implications of channel mobility, the role of PIEZO1

in cell migration, and molecular and biophysical mechanisms associated with PIEZO1 localization and activation in neural development, to name a few. We are very excited about the potential for the PIEZO1-HaloTag platform to facilitate rigorous, quantitative, and mechanistic studies addressing these questions. However, we believe that these studies are beyond the scope of the present work, and deserve dedicated, independent investigations.

Overall, we believe the additions to our manuscript have highlighted the utility and versatility of the PIEZO1-HaloTag line to unravel PIEZO1-mediated biology. The primary focus of this study is to rigorously validate the PIEZO1-HaloTag system and to establish its ability to quantitatively track localization and activity of endogenous PIEZO1 under native cellular conditions with high precision. We believe that our novel platform presents a significant advance over existing approaches for studying human PIEZO1 and complements other approaches to study PIEZO1, advancing both physiologically- and clinically-relevant research on human PIEZO1. In follow-up studies, we are excited to employ this tool to uncover novel biology related to PIEZO1.

6. Throughout the manuscript, the authors note that this new PIEZO1-Halotag technology is a significant improvement over previously existing approaches using PIEZO1-tdTomato. This claim would be greatly improved if the authors included direct comparisons to PIEZO1-tdTomato that showed statistically significant improvements to studying PIEZO1. There are a few sections throughout the manuscript where additional statistical comparisons would be helpful:

- a. In Figure 2A, is this increased photostability in the Piezo1-HaloTag puncta significantly greater than PIEZO1-tdTomato? The differences shown in the graph were noted, but there is no statistical test comparing the two groups.
- b. In Figure 2B, the authors show PIEZO1-HaloTag in a migrating neural stem cell. In the text, this is said to show improved image quality over PIEZO1-tdTomato, but this is not explicitly shown in the manuscript. Nor is there any statistical analysis to support significant improvement.

We thank the reviewer for highlighting the need for additional statistical comparisons. We have added further statistical comparisons throughout the work. We found the difference in the time constant of bleaching between PIEZO1-tdTomato and PIEZO1-HaloTag samples had a Mann-Whitney p-value of 5.29×10^{-7} and a Cohen's *d* value of -2.73, indicating the samples are significantly different with large effect size (Fig. 2A).

In Fig. 2B we now report the localization error, i.e. the deviation from the mean position of a trajectory, for PIEZO1-tdTomato and JF549 labeled PIEZO1-HaloTag in endothelial cells. PIEZO1-tdTomato has a localization error of $52 \pm < 1$ nm while JF549 labeled PIEZO1-HaloTag samples has a significantly improved localization precision of $33 \pm < 1$ nm (Fig. 2B, lines 149-153).

Finally, we calculated the signal-to-noise ratio of both PIEZO1-tdTomato and PIEZO1-HaloTag samples, noting a large improvement in the signal-to-noise ratio in JF646 labeled PIEZO1-HaloTag samples relative to PIEZO1-tdTomato (5.62 ± 0.19 vs. 3.14 ± 0.12) (line 148-149).

Together, these results highlight the superior image quality and precision of the PIEZO1-HaloTag system, demonstrating its significant advantages for studying PIEZO1 dynamics at the molecular level.

7. In Figure 3E & 3H, there is a noted increase in the occupancy in active state of the Yoda1 treated group compared to the DMSO treated group, but this finding could be further strengthened with a statistical comparison to show the significance of this result.

We thank the reviewer for this comment. In the revised manuscript we present additional quantitation of the data (Fig. 3H caption, line 250-251). Specifically, we conducted two two-sample

Kolmogorov-Smirnov tests to determine if the Yoda1 distribution was significantly different from both the Untreated and the DMSO distribution. Both tests returned a probability value lower than 0.001, indicating that the Yoda1 distribution is significantly different. We conducted the same analysis for data comparing channel activity between ECs and NSCs (Fig. 4G, line 354-355); the two distributions were also significantly different.

Minor Comments:

1. Figure 1F – Including labeling in the figure that these images are showing PIEZO1-HaloTag labeled with JF646 would benefit the reader.

We have added additional labeling to Figure panels 1E and 1F in order to clarify the images are using JF646 (page 3).

2. Figure 1D – The authors can statistically show that P1 KO significantly reduces electrical current following mechanical stimulation. Currently the authors do not show a statistical comparison of P1 WT and P1-HT. Including the statistical results for the comparison between P1 WT and P1-HT would be nice to show that P1-HT isn't significantly altering the channel activity.

We thank the reviewer for this comment. We have now added to the legend for Figure 1D the p-value and Cohen's *d* value comparing the current amplitudes from PIEZO1 and PIEZO1-HaloTag conditions. The addition of the HaloTag does not significantly alter channel activity when compared to PIEZO1 WT. In addition we show in Supplementary Figure 4 that attachment of a HaloTag ligand doesn't alter the functionality of the PIEZO1-HaloTag channel (lines 125-131).

3. Figure 2A – Including a label of what the red and grey lines represent in the graph would benefit the reader.

We have now included labels of the red and grey lines to Figure Panel 2A. Additionally, we revised the associated figure legend for clarity "PIEZO1-HaloTag puncta are brighter and bleach slower than PIEZO1-tdTomato puncta. TIRF image series of PIEZO1-HaloTag and PIEZO1-tdTomato endothelial cells were acquired for 2 minutes with identical acquisition settings (see Methods). Red trace indicates the average integrated intensity of PIEZO1-HaloTag ($n = 19$ videos from 3 independent experiments) and grey trace indicates the average integrated intensity of PIEZO1-tdTomato ($n = 20$ videos from 3 independent experiments). The signals are scaled to the number of puncta in the first frame for each video. Black dashed curves represent exponential fits to the data with $\tau_{HT} = 38.1 \pm 0.15$ s and $\tau_{tdT} = 21.5 \pm 0.13$ s (p-value Mann-Whitney < 0.005 and the Cohen's *d* value = -2.73)" (Fig. 2A, page 5).

4. Referencing the supplemental videos in the text would be another benefit to readers. Currently supplemental videos are mostly cited in figure legends. For example, referencing supplemental video 4 on page 6 would give the readers an example of mobile puncta.

We have added a reference to Supplemental Video 4 with our reference to Figure 2D. We additionally confirmed that all supplemental figures and videos are now referenced throughout the manuscript.

5. Figure 4G – Rather than writing "Same as E but..." in the figure legend, explicitly stating the information would be helpful to readers. Additionally labeling on the graph to note that this with JF646-BAPTA HTL would also be helpful for readers.

We have clarified the legend for Figure 4G (now Figure 5G) to “Density scatter plot of distances of JF646-BAPTA HTL labeled active PIEZO1-HaloTag puncta localizations to the lumen edge mask (X axis) and to the outer edge mask (Y axis) of the MNR (n = 39 videos from 12 MNRs from 3 experiments). The color scale indicates the relative density of puncta at each position in the scatter plot, scaled to the total number of puncta represented in the plot. Active channels localize primarily near the actin-rich lumen, with a smaller cluster near the actin-rich outer edge of the MNR. Density scatter plots for each individual JF646-BAPTA HTL sample can be found in Supplemental Fig. 18. See also Supplemental Figs. 16, 17, 19, and 20 and Supplemental Video 8.

Response to Reviewer #3:

Bertaccini et al.'s manuscript is an interesting integration of the HaloTag method for visualising, tracking PIEZO1 in cultured cells as well as neuronal organoids, and for tracking local calcium. It is a nice demonstration of the principle of this approach, however pivots heavily on a couple of assumptions that need validating. I have proposed a number of major points that I think need to be addressed before I would recommend this for publication.

Major:

1. Page 4, paragraph 2: The JF HaloTag benchmarking against the tdTomato is fairly convincing. However, JF dyes are primarily designed for photoswitching experiments, hence the photobleaching can be reversible. For them to be used in experiments showing recovery of renewal of PIEZO1, authors will need to demonstrate the extent of the non-recovery from the dark state (after a dark/non-excited period). Until then, it is impossible to discern this fluorescence recovery (e.g. Figure 2b) from the recovery of JF646 from their triplet/dark states.

We thank the reviewer for their comment. First, we would like to clarify that we do not perform FRAP measurements (where recovery of renewal of PIEZO1 signal would be relevant) in our study. Second, whereas Janelia Fluor dyes are inherently brighter and more photostable than traditional fluorescent proteins, not all JF dyes are designed to be photoswitchable or photoactivatable. The HaloTag JF dyes including JF 646 and JF 549 are not photoswitchable (Grimm et al. 2015, PMID: 25599551), but they are fluorogenic, i.e. they increase in their fluorescence emission intensity when attached to the HaloTag domain. Photoswitchable JF dyes have also been developed for super-resolution imaging (e.g. Photoactivatable JF549 and JF646 (PA-JF549, PA-JF646)) (Grimm et al. 2016, PMID: 27776112), but we do not use these in our study. We have only used the non-photoswitchable JF549, JF635, and JF646-based dyes.

We realize that unclear referencing on our part may have led to this confusion, i.e. we referenced in the introduction, the Photoactivatable JF dyes (Grimm et al. 2016, PMID: 27776112) "Combined with the use of bright and photostable Janelia Fluor (JF)-based HaloTag ligands³⁷⁻³⁹, super-resolution imaging, and automated image analysis, our approach allows the study of endogenous human PIEZO1 through microscopy assays in various hiPSC-derived cell types...". For clarity, we have removed the reference to the photoactivatable dyes (Grimm et al. 2016, PMID: 27776112) from this sentence. Lines 63-66.

2. Page 6, paragraph 2: I am surprised that even after fixation, the authors' tracking approach detects a mean path length peaking around 1.2 micron - particularly as the localisation uncertainty is 20 nm. I suspect the 3-pixel tracking radius criterion is too broad. I would like to see this tracking protocol benchmarked and optimised against image sequences of completely immobilised (in vitro, and drift corrected) single dye ligands. Until then, I am not convinced about the (particularly large) track lengths and diffusion co-efficients derived for PIEZO1 here.

We localized puncta in successive image frames and determined the localization precision by measuring the distance of each localization from the mean position throughout records from fixed samples. The localization precision was $33 \pm < 1$ nm for PIEZO1-HaloTag samples labeled with JF549, $29 \pm < 1$ nm for PIEZO1-HaloTag samples labeled with JF646, and $52 \pm < 1$ nm for PIEZO1-tdTomato puncta (line 149-153). Over the course of 5 seconds we thus expect that the mean path length (sum of localization errors) would be about 1.45 micron ($5 \text{ s} \times 10 \text{ frames per second} \times 29 \text{ nm}$). This is consistent with the ~1.5 micron mean path length observed for our fixed sample data.

In order to address localization errors that may arise from x-y drift in our samples, we imported the localizations of each extracted trajectory into ThunderSTORM. Utilizing the drift correction

module in ThunderSTORM we determined that the drift in our PIEZO1-HaloTag fixed sample data was negligibly small (< 0.1 pixels, i.e. 10 nm, over 5 s). This is now reported in a new methods section “Evaluation x-y Drift in Immobilized PIEZO1-HaloTag.” lines 727-736

3. The data presented in Figure 2, in my view, are an over-simplification of the PIEZO1 tracking outcomes.

- a. Firstly, the authors use linear fits to the first couple of seconds of the MSD/t curves to obtain the diffusion coefficients. However those curves are clearly non-linear, which to me suggests that a straightforward passive diffusion process is not likely to explain the PIEZO1 mobility. It is possible that there is a cumulative effect of drift in the sample which does not seem to have been addressed explicitly. Assuming that drift was negligible, the non-linear nature of these curves indicates a subdiffusive movement that has neither been explored nor ruled out in the paper.

We have addressed drift, as described previously in our previous response, by using the drift module in ThunderSTORM and noting a negligible contribution of drift.

We acknowledge the reviewer’s comment that our curves are clearly non-linear, an observation that was reported in our description of these results “At longer times the relationship fell below linear (Fig. 2H), indicating sub-Brownian, anomalous diffusion”. Additionally, we note that the fit to the initial two seconds of the mean squared displacement represents the upper limit to the diffusion coefficient, and we have clarified this in lines 218-220 stating “A linear fit up to 2 s for the immotile population, representing the upper limit of the diffusion coefficient, yielded an apparent diffusion coefficient of $0.029 \mu\text{m}^2/\text{s}$, representing the upper limit of the diffusion coefficient”

- b. Secondly, the multimodal Gaussian curve fit approach to the path lengths histogram is not particularly convincing based on the data provided here. Even for a passive Levy walk, typical track lengths of 4-6 microns over just 5 seconds for a macromolecular complex such as PIEZO1 is fairly high. The authors provide no reference data to PIEZO1 diffusion in ‘clean’ environments such as reconstituted bilayers. Given the lack of other validating information such as drift correction and event-specific localisation error, I find this data inconclusive.

Currently, there is no data available for PIEZO1 diffusion in reconstituted bilayers. The diffusion coefficient for another mechanically activated ion channel, human TRAAK, reconstituted in a freestanding bilayer was reported as $D=0.8\mu\text{m}^2/\text{s}$, $\alpha=0.92$ (Pérez-Mitta et al. 2024, PMID: 38905330). For PIEZO1, the Mackinnon lab has reported the diffusion coefficient of PIEZO1 in red blood cells, which do not possess organelles or other transcytosolic networks (Vaisey et al. 2024, PMID: 36515266). Their diffusion coefficient from the average MSD of the initial 50 milliseconds of PIEZO1 mobile trajectories pre confinement is $0.037 \mu\text{m}^2 \text{s}^{-1}$, similar to our reported value of $0.029 \mu\text{m}^2 \text{s}^{-1}$. We also previously reported the apparent diffusion coefficient of PIEZO1 in neural stem cells (Ellefsen et al. 2019, PMID: 31396578) as $0.067 \mu\text{m}^2 \text{s}^{-1}$ and of IP3R as $0.064 \mu\text{m}^2 \text{s}^{-1}$ (Smith et al. 2014, PMID: 25140418). Thus our value is within the reported range for large ion channels. This reference data has been added (lines 221-224).

We have addressed drift, as described previously in our response (Reviewer 3, comment 2b), noting a negligible contribution of drift.

4. Page 8, paragraph 1: The calcium time traces in the experiment using JF646-BAPTA resemble single channel opening events, however photo-electron calibrated intensity units would be more helpful in discerning whether these are photon-limited events representing single-PIEZO1 openings or indeed small clusters. As a minimum, data like these should be accompanied by an in-vitro calibration of the single molecule (single JF646_BAPTA HaloTags) fluorescence in response to controlled calcium levels which will indicate whether the single molecule-level of calcium sensing is achievable in the imaging conditions used here. Given that JF646 is inherently photoswitchable, the authors have not additionally shown any evidence to eliminate the possibility (or a component) of blinking or radial fluctuations in these fluorescence traces. The supplementary movies included are so saturated that it is impossible to tell how such stepwise fluorescence traces are obtained.

We thank the reviewer for their suggestion. We have now included conversion factor from camera units (c.u.) to photoelectron counts in the “Total Internal Reflection Fluorescence (TIRF) microscopy Imaging” Methods section (page 18, lines 35-36).

We also thank the reviewer for their suggestion to calibrate our single molecule JF646 BAPTA HaloTag measurements. To do so, we transfected WTC-11 hiPSCs with plasmid expressing a monomeric cytoplasmic HaloTag protein (pCDNA5-FRT-TO_HaloTag7_T2A_EGFP Addgene plasmid#: 169325) (Frei et al. 2022, PMID: 34916672) and labeled transfected cells with JF646-BAPTA HTL and JF549 HTL (as previously mentioned, JF549 and JF646 are not inherently photoswitchable, but are fluorogenic). Then, we fixed and permeabilized labeled cells and imaged them in 0 nM, 38 nM, 100 nM, 225 nM, 602 nM, and 39 μ M Ca^{2+} solutions at 23C.

Fluorescence intensity of the non- Ca^{2+} -sensitive JF549 puncta plotted over time remained steady, as expected (Supplemental Fig. 9A). On the other hand, JF646-BAPTA puncta were brighter in the presence Ca^{2+} . We show example intensity traces of fixed BAPTA-HTL in each Ca^{2+} concentration. In the absence of Ca^{2+} , fluorescence intensity of JF646-BAPTA-labeled puncta was very dim, just above background. Moreover, the fluorescence traces from individual puncta displayed flickering behavior which increased in frequency as Ca^{2+} concentration increased. At the highest Ca^{2+} concentration, puncta frequently spent extended periods of time in the bright state with some puncta continuously bright. The data demonstrates that under the imaging conditions used, signals recorded from the PIEZO1-HaloTag JF646-BAPTA labeled cells indeed arise from Ca^{2+} binding to the JF646-BAPTA HTL. This data is shown in Supplementary Figure 6 and discussed in the manuscript on lines 125-131.

5. Page 19, paragraph 3 “In order to remove any spurious detections due to unbound JF635 dye passing through the plane of imaging...”: Is there a specific reason why the excess JF635 Halo ligands were not washed out during imaging? Whilst it has been shown by the Lavis group that the fluorescence is up to 80-fold dimmer when the ligand is unbound, this would still make a non-zero contribution to the background. Whilst the authors go through an elaborate time-based filtering of background fluctuations, I would like to see how much of these spurious detections have been due to the practice of not washing out the excess probe. How much of the background observed in the PIEZO1-HaloTag KO samples was caused by this approach?

We would like to clarify that all samples were treated with JF635 overnight and washed subsequently 3 times with culture media prior to fixation or live imaging as described in Methods subsection “HaloTag Ligand Treatment Protocol” (lines 542-552). Despite washing out excess dye, it is possible that some unbound probe is present within multiple cell layers of the MNR samples. This is demonstrated by examination of our PIEZO1-HaloTag knockout control samples that have been labeled with JF635 (Supp. Fig. 19) – even after wash steps, some signal not attributed to

autofluorescence spots is seen – we note this as “any spurious detections due to unbound JF635 dye”.

For clarity, we have now included more detail on our filtering methods and in the Supplementary Results section (see edits on pages 33-34), and also added definitions of the three kinds of spurious signals observed in our samples (lines 189-192). We have also modified Supp. Fig. 19-20 to include quantitation of the number of puncta remaining after each filtering step (Supp. Fig. 19 and 20, page 35-37).

Minor:

1. Page 2, The orthogonal assays of positive expression of PIEZO1-HaloTag and PIEZO1 HaloTag KO are compelling, however the protocols of cell lysis and immunoblot have not been detailed. For full transparency, this information needs to be included either in the main text or supplement.

We thank the reviewer for bringing this inadvertent omission to our attention. We have now added this information to the Methods subsection titled “Cell Lysis and Immunoblotting” (lines 628-642).

2. Page 4, paragraph 2: When reporting the protocols for photobleaching experiments, it is imperative that you report the excitation laser power as a function (e.g. in W/cm²). This becomes more crucial when experiments showing so-called single-channel activity are shown with the same imaging probes.

We have now reported the excitation laser power as a function (W/cm²) in the TIRF Methods section ((page 18, lines 27-28 and 31-33).).

3. Final paragraph of the Results: Authors use 77 photons as the threshold point (based on negative controls). However, there is no information provided on the sample-to-sample (and depth dependent) variation in the background. I find this 'computational' thresholding protocol no more arbitrary than manual eyeballing. This needs to be discussed as a key limitations of the approach.

To demonstrate our rationale for the threshold of 77 photons selected, we have added histograms showing the relative frequency of intensity values of puncta from representative samples of JF646-BAPTA PIEZO1-HaloTag and controls JF646-BAPTA PIEZO1-HaloTag KO and unlabeled PIEZO1-HaloTag MNRs (Supp. Fig. 20F). Based on these histograms, the threshold was empirically determined and set at the 70th percentile of the detected object intensities in the JF646-BAPTA PIEZO1-HaloTag KO condition, corresponding to approximately 77 photons. This threshold provides a reasonable balance between removing too many bona fide PIEZO1 puncta and retaining too many spurious detections.

We have revised the supplementary results section for greater clarity and detail and included a sentence discussing how the intensity-based filtering does not remove as many of the spurious detections as the time-based filtering does (line 251-253).

4. Paragraph 3, page 17: Multiple typographic errors ‘rois’, need to be revised to ‘ROIs’.

We thank the reviewer for identifying these typographic errors. We have corrected ‘rois’ to ‘ROIs’ in paragraph 3 page 17 (now line 751-760).

RESPONSE TO REVIEWER COMMENTS

Reviewer #1:

Great work! The authors have addressed all my comments and questions. I don't have any further comments.

We thank the reviewer for the kind words, and for their inputs which significantly strengthened the manuscript.

Reviewer #2:

1. In this revised manuscript, Bertaccini et. al. have added depth to their study showcasing the utility of their newly developed PIEZO1-HaloTag system as a way of studying endogenous Piezo1 activity. The additional statistical tests, supplementary figures, and additions in figure 4 address many of my concerns from the original submission. I only have a few comments that remain.

Line 127, "...mechanical stimulus imparted by with poking using a blunt glass probe (poking assay)." This phrasing is a little awkward. The awkwardness might be resolved by removing "by".

We thank the reviewer for their comment. We have fixed the typographic error in this statement. It now reads "...mechanical stimulus imparted with poking using a blunt glass probe (poking assay)." (Line 125-127, page 4).

Figure 3 D, F, and I include expanded tracings in red to the right of the original graph. The time scale on these goes from 0 seconds to 3 seconds (D and F) or 0.5 seconds (I). Because these expanded tracings correspond to the red marked regions on the left graph, should the time scale not also reflect the corresponding time? The authors seemingly did this in Figure 4E, although the timescale on the expanded tracing only fits the neural stem cell graph.

We thank the reviewer for highlighting the change in axis in the expanded traces in Figure 3D, F, and I. In Figures 3D and 3F, we decided to restart the x axis origin from time 0 s so that the same axis could apply to all three of the above traces. Otherwise, the x axis would not accurately represent all three traces together. However, we do recognize that this labeling strategy was not used in Figure 4E, and we have edited this axis to also begin at 0. Furthermore, the x axis in Figure 3I does not need to restart from 0, as we are not illustrating multiple traces. We have edited this panel to the reviewer's suggestion with the x axis showing where in the recording the traces are occurring. (Figure 3D, F, and I, page 7; Figure 4E, page 10).

Aside from differences in PIEZO1 expression levels, do the authors have any additional insights as to why NSC's have fewer and less active puncta than ECs? These results showcase the utility of your new tool, but is there additional significance or importance to these findings? Showing the significance or importance of this data will in turn show the impact of this study and tool on the field.

We thank the reviewer for their question. We have now added a sentence to the discussion "The PIEZO1-HaloTag hiPSC also allows for the comparison of relative differences in PIEZO1 activity between distinct cell types. NSCs display less active PIEZO1 puncta than ECs which we speculate may be due to differences in cell type-specific contractility that activates PIEZO1³⁰. We also provide proof-of-principle that the PIEZO1-HaloTag platform allows measurement of PIEZO1 activity in response to externally-applied mechanical stimuli such as osmotic stimulus. These activity measurements can be extended to other mechanical stimulus modalities in the future." (Lines 491-496) addressing this comment. Rigorously investigating the mechanistic reason why NSCs have fewer and less active puncta than ECs is an exciting

question that we hope to answer in future work. In the meantime, we have also added an additional figure (Supplementary Fig. 6) illustrating that – like ECs – NSCs have both populations of mobile and immobile PIEZO1-HaloTag puncta, and a similar diffusion coefficient.

In Figure 4A, the area surrounding the NSC has several puncta that are a similar size and brightness to those noted inside of the cell. Are these puncta a result of high background intensity or are these puncta from surrounding cells?

We thank the reviewer for their question. These puncta are outside the cell on the coverglass. As shown in the “Table of HaloTag Ligands,” we perform overnight treatment with HaloTag ligands. These puncta may result from non-specific binding of the probe to the glass, or from debris left outside the cell during migration or from labeled cells detaching from the coverglass during washing steps, leaving a smaller residual signal.

Lines 327-341, Using PIEZO1-Halotag to investigate PIEZO1 response to osmotic stress is an innovative approach that showcases more utility for this developed tool. The methodology for changing osmolarity is currently hard to find in the Methods section. Under the Total Internal Reflection Fluorescence (TIRF) microscopy Imaging subheading there is information on the solution with osmolarity of 316 mOsm/L. I currently do not see information on the 251 mOsm/L hypotonic solution in the methods section.

We thank the reviewer for identifying this omission. We have now added information to the Methods section (“Total Internal Reflection Fluorescence (TIRF) Microscopy Imaging”) regarding the 251 mOsm/L hypotonic solution (Lines 717-718, page 19).

Reviewer #4:

Bertaccini et al. utilized human induced pluripotent stem cells (hiPSCs), CRISPR/Cas9 genome editing, and the well-established HaloTag technology to develop advanced imaging and analysis workflows for investigating the mechanosensitive PIEZO1 channel. Their study focused on PIEZO1 localization, diffusion dynamics, and channel gating using calcium sensors. This approach was demonstrated not only in different differentiated single cell types but also in a living neural organoid model using advanced light-sheet imaging. They successfully imaged individual PIEZO1 channels (punctae), analyzed their mobility and diffusion through single-particle tracking, and demonstrated the superiority of their method over previous studies based on the fluorescent protein tdTomato. Notably, the application of Janelia Fluor JF646-BAPTA-3'AM probes for high-frame-rate calcium imaging represents a significant contribution to the manuscript.

The authors have made significant improvements to the first version of the manuscript, addressing and clarifying most of the original concerns raised by the former reviewer #3. I am in favor of publication but would like to suggest a few additional minor modifications. Below are my point-by-point comments on the authors' responses to the former reviewer:

We very much appreciate the reviewer stepping in to evaluate our manuscript and providing constructive feedback.

(1) I agree that this was likely a misunderstanding by the reviewer and the mentioned references concerning the JF dyes are correct.

(2) Using the total path length over a defined time period (here, 5 seconds) as a criterion for identifying immobile particles, while valid, may have led to a misunderstanding by the previous reviewer. It could also be challenging for readers without a deeper understanding of single-particle tracking to interpret. The total path length of an immobile particle increases monotonically at each step due to localization errors.

Moreover, presenting the cumulative histogram of squared displacements (Fig. 2F) does not effectively highlight the presence of immobile particles. Sample drift was correctly addressed by the authors and appears to be negligible.

First, I suggest including a movie of the raw data with overlaid trajectories within a smaller ROI, such as a portion of the cell shown in Supplementary Movie 4 (Fig. 2C), to demonstrate that the tracking works effectively.

We thank the reviewer for their suggestion. We have now added Supplementary Video 5 showing our raw data overlaid with trajectories for a zoomed-in region of a PIEZO1-HaloTag endothelial cell, illustrating trajectories generated for the analysis in Fig. 2F-I.

Additionally, I recommend presenting a histogram of single-step length distributions (neither squared nor cumulative) and highlighting the region corresponding to localization precision, which should show a distinct peak representing immobile trajectories.

Moreover, MSD analysis is based on single trajectories, as shown in Fig. 2I, is widely used in the field to differentiate between mobile and immobile particles. I suggest retaining Figs. 2F and 2I while adding the single-step length distribution plot. Figs. 2G and 2I could be moved to the supplementary information to keep the criterion for immobile punctae.

We thank the reviewer for their recommendation regarding plotting single SLDs, as this was the initial strategy we tried to highlight the differences between PIEZO1-HaloTag immobile and mobile trajectories. We first plotted the histograms of the single-step length distributions (neither squared nor cumulative) for our fixed sample data, representing our immobile tracks. In the histogram below shown in black, we only observed a single broadly distributed peak. When we plotted the same for live sample data, shown in the plot below in red, we observed only a slight increase in the spread of the single-step length distributions, which was not as distinct as when we plotted the cumulative step length distributions over 5s in Figs. 2G-H. We believe this is because PIEZO1 is moving slowly enough that the single step length distribution does not highlight the difference in mobile versus immobile motion as much as cumulative path length.

Figure 1. PIEZO1-HaloTag endothelial cell fixed and live trajectory single-step length distributions. PIEZO1-HaloTag endothelial cells labeled with JF646 HTL were imaged Fixed (black, number of tracks = 14,943 from 13 videos) or live (Red, number of tracks = 889,971 from 40 videos) at 10 fps. Single-step length distributions from extracted trajectories are shown as histograms.

Finally, the authors should discuss the fraction of nonspecific binding of HaloTag ligands to the cell surface, which could contribute to immobile signals. Referring to Supplementary Movies 2-4, there appears to be a significant fraction of signals on the glass surface outside the cell area.

To address the reviewer comment, we labeled PIEZO1-HaloTag WT and KO endothelial cells with JF646 HaloTag ligand, and imaged using our standard imaging methodology. We then analyzed both PIEZO1 WT and KO labeled samples using the trajectory extraction protocol described in the methods section “Analysis of PIEZO1-HaloTag Puncta Diffusion based on TIRF imaging.” Here, analyzing $n = 6$ videos from 3 biological replicates we find that there is very little contribution of spurious tracked puncta. The PIEZO1-HaloTag samples had 0.5 ± 0.05 tracked puncta per μm^2 while the PIEZO1-HaloTag KO samples had 0.004 ± 0.0006 tracked puncta per μm^2 (** p -value < 0.005 , Cohen’s d effect size = -5.44). The number of tracked puncta per μm^2 have been added to the main manuscript text (Lines 195-199, page 6).

Figure 2. PIEZO1-HaloTag track extraction captures few spurious trajectories.

PIEZO1-HaloTag WT and KO endothelial cells were labeled with JF646 and puncta were tracked as described in the methods. The number of tracked puncta per area of the cell were counted. WT: Blue, 0.5 ± 0.05 tracks per μm^2 and KO: Orange, 0.004 ± 0.0006 tracks per μm^2 , ** p -value < 0.005 , Cohen’s d effect size = -5.44.

(3) a) Using linear fits over the first few seconds of the MSD is common in the field for determining apparent diffusion coefficients. The results are in agreement with the diffusion coefficients of protein complexes in the plasma membrane of living cells.

We thank the reviewer for this comment.

b) I believe this likely stems from a misunderstanding, where the total track length is confused with the end-to-end trajectory distance, which is significantly shorter in the case of a simple random walk. The authors have appropriately added references to support their findings regarding the measured diffusion coefficients

We thank the reviewer for this comment.

(4) Regarding this point raised by the previous reviewer, I have a general question about the data interpretation. In the Methods section, the authors stated that all samples were labeled with only 500 pM at 37°C. However, the incubation time, which should be provided in a table (cp. line 648) is missing. At such low concentrations and with a reasonable incubation time (e.g., 30 minutes), one would expect that only a small fraction ($\leq 10\%$) of HaloTags would be labeled (cp. T. Appelhans et al., Nano Lett. 2012, 12, 610–616). Although the PIEZO1 channel is a trimer, most channels are likely labeled with only a single dye under these conditions. This should be discussed or clarified and could potentially be determined by analyzing single-puncta bleaching events (as e.g. Hummert J et al., Mol Biol Cell. 2021 Nov 1;32(21):ar35. doi: 10.1091/mbc.E20-09-0568 or Danial et al., The journal of physical chemistry letters, 13(3), 822–829. <https://doi.org/10.1021/acs.jpcclett.1c03835>) in time-lapse recordings, which should already be available and could be reviewed. What was the rationale behind the low labeling concentration?

We thank the reviewer for raising this important question and apologise for missing the incubation times in the Methods. We have now added both incubation time and concentration columns to Supplementary Methods Table No. 1. This table now shows that for our Ca²⁺-insensitive dyes that we perform an overnight labeling at 500 pM, which is expected to label most of the puncta. We performed many troubleshooting experiments which illustrated that higher concentrations of dye led to significant background outside the cell and on the substrate. On the other hand, at lower concentrations, we observed sparser puncta, suggesting that not all puncta were getting labeled. For the Ca²⁺-sensitive dye, when we increased labeling concentration or the incubation time, we observed punctate signal that went in and out of the TIRF field (i.e. in the z-dimension); we attributed this to excessive (unbound) probe being trapped within the cell after AM-ester cleavage, ie. over-labeling of our samples.

I have another general question regarding the Ca²⁺ measurements. The authors state that the BAPTA probe can be used to analyze single-channel gating through direct analysis of intensity fluctuations (lines 262–264 and 297–300). However, the rate of Ca²⁺ chelation and de-chelation should also be considered as a potential rate-limiting step. In particular, the off-rate (de-chelation) could be significantly slower than the closing of the channel.

We thank the reviewer for their comment. We acknowledge the rate of Ca²⁺ chelation and de-chelation is an important consideration for the interpretation of the JF646-BAPTA bright state estimations. We have now added a sentence in the discussion section “However, the observed “bright state” durations may overestimate channel open times as a consequence of the K_{off} of the HTL and the time course of Ca²⁺ diffusion away from the pore.” (Lines 477-479, page 13). Additionally, since the manuscript resubmission, we identified an error in the calculation of our Ca²⁺ Calibration buffer solution concentrations. Following corrections, we have retained the data for 0 nM, 75 nM, and 39 uM conditions.

Additionally, examining the intensity traces in Fig. 3D, Fig. 3F, and Supplementary Figures 6–14, a single intensity level - and occasionally two levels - are visible most of the time. This suggests that most channels are labeled with only a single dye. The labeling efficiency and its implications for data interpretation should be more explicitly addressed in the Results and Discussion sections.

We thank the reviewer for highlighting the labeling efficiency as a crucial discussion point. We have now added further interpretation to the discussion section regarding the efficiency of HaloTag labeling and its implications for interpretation of the JF646 BAPTA HTL intensities. The paragraph now reads (new/edited text in maroon font (Lines 471-484, page 13).

“Our findings advance existing methodologies, such as GenEPI⁶⁵, a genetically-encoded fluorescent reporter of PIEZO1 activity, which relies on overexpression and is limited by poor kinetics and weaker signals. Activity measurements from PIEZO1-HaloTag revealed multiple levels of JF646-BAPTA HTL fluorescence intensity from single, stationary puncta. Although saturated labeling by the probe involves three HaloTag ligands per PIEZO1 trimer, the high [Ca²⁺] in the immediate proximity of a channel pore, estimated at > 15 μM⁵⁴, implies that the three HTLs would respond almost simultaneously to Ca²⁺ flux, given their proximity to the channel pore and high affinity for Ca²⁺ (K_d 0.14 μM)⁴⁴. However, the observed “bright state” durations may overestimate channel open times as a consequence of the time courses of Ca²⁺ unbinding from the HTL and of Ca²⁺ diffusion away from the pore. We speculate that multiple levels of JF646 BAPTA HTL fluorescence may result if each diffraction-limited punctum represents a cluster of two or more PIEZO1 channels. Differences in amplitudes of signal from different PIEZO1-HaloTag puncta could thus arise from differences in number of PIEZO1 channels per punctum, or due to varying distances of the membrane within the exponentially decaying evanescent field, or sub-saturation labeling of some channels. Importantly, recent super-resolution studies using MINFLUX^{56,66} imaging of PIEZO1 demonstrate clustering of PIEZO1⁶⁷. ”

(5) Data and Code Availability: Two GitHub repositories are listed and accessible. However, the first repository (<https://github.com/Pathak-Lab/PIEZO1-LocalizationTools>) lacks a minimal example with raw data to test the code, and no hardware requirements or installation instructions are provided. The second repository (https://github.com/abcucberkeley/piezo1_analysis_pipeline) includes a minimal example, but hardware requirements are also missing. It appears that a high-performance workstation is needed, as I was unable to run the code on a smaller workstation."

We thank the reviewer for identifying these omissions. We have now updated the PIEZO1-Localization Tools repository with a minimal example with raw data to test the code. In the same repository, we have now provided hardware requirements and installation instructions. The second GitHub repository now includes the hardware requirements.

Minor concerns:

Minor concerns (1) and (3–4) raised by the former reviewer were adequately addressed by the authors. I have one suggestion for point (3):

The authors reported the laser power in units of $[W/cm^2]$ at the back pupil plane of the objective. However, these units represent power density, not power, and should instead be estimated for the focal plane of the objective. Typical values range from 10 to 300 W/cm^2 , depending on the exposure time (1–30 ms). Alternatively, the authors could report the power in $[W]$ at the back pupil plane, as they do in the AO-LLSM section.

We thank the reviewer for their suggestion and for catching that here we were in fact reporting power density, not power. We have followed the reviewer's feedback and updated these values to their corresponding power in $[W]$ at the back pupil plane of the objective (Lines 724-730, page 19).

Here are some additional minor points that the authors should address:

(1) Fig. 1B: Please add labels and a legend to the structure, along with a color code.

We thank the reviewer for their suggestion. We added to our existing caption the color coding for each aspect of the PIEZO1-HaloTag structure located in Figure 1B. Additionally, we have added labels directly in the Figure 1B panel itself to directly identify each component of PIEZO1-HaloTag (Figure 1B, page 3).

(2) Terminology: Throughout the text, the authors use both "mobile vs. immobile" and "motile vs. immotile." I recommend using consistent terminology and sticking to one set of terms.

We thank the reviewer for their recommendation. We have now changed "motile" and "immotile" to "mobile" and "immobile" throughout the manuscript, ensuring consistent terminology.

(3) MSD Analysis in Fig. 2I: I suggest using the term "lag time" for the axis label in the figure and in the text to distinguish lag time from the simple time points of the acquisitions.

We thank the reviewer for their suggestion. However, the MSD we are plotting here is referring to the time point of acquisition. As such, we believe it is accurate to leave the axis label as time (s).

(4) Figure 3: This figure is supported by three movies showing very small ROIs with flickering of the JF646-BAPTA HTL probe. I would like to see a movie with a short sequence of the full cell at a high frame rate (200 fps) to illustrate the flickering of the BAPTA probe in single-cell level.

We thank the reviewer for their comment. In order to acquire videos with a temporal resolution of 200 fps with our camera settings, we must reduce the field of view to a 52x52 pixel size as shown in our supplementary videos. To address the reviewer comment, we have added a supplementary video (Supplementary Video 6) illustrating a full cell of PIEZO1-HaloTag JF646-BAPTA labeled endothelial cells

in Basal and 2 μ M Yoda1 conditions acquired at 100 fps for a duration of 10 seconds. This video uses the same data as shown in Figure 3A with a 512x512 pixel roi.

(5) Methods Section (line 648): A Supplementary Methods Table No. 1 is mentioned, which should list the incubation times for all JF-HTL dyes, but this information is missing.

We thank the reviewer for catching this omission. We have now clarified that Supplementary Methods Table No. 1 is referring to our “Table of HaloTag Ligands”, and cited this table in the Methods section titled “HaloTag Ligand Treatment Protocol”. We have edited the “Table of HaloTag Ligands” to include the concentration and incubation time used for each of the JF-HTL dyes (Lines 947-948, page 25).

(6) Suppl. Fig. 1: Please add a legend for abbreviations in the figure caption.

We thank the reviewer for their suggestion. We have added the legend for relevant abbreviations to the figure caption (Lines 2-7, Supplementals page 1).

(7) Suppl. Fig. 6: The figure is incomplete/clipped and needs to be corrected.

We thank the reviewer for catching this error. We have now updated Supplementary Fig. 6 (Lines 44-54, Supplementals page 6).

(8) Suppl. Information File (line 243): The authors give the threshold in photons here. I suggest sticking to camera units (c.u.) throughout the manuscript or converting all camera units to photons using the given conversion factor.

We thank the reviewer for their suggestion. For clarity, we have added the corresponding value of 77 photons (261 c.u.) in the main manuscript (Line 884, page 23) as well as in the Supplementary Methods (Line 310, Supplementals page 39).

(9) Suppl. Figs. 19 & 20: The detections (green circles) in panels B, C, and D are very small and should be adjusted for better visibility.

We thank the reviewer for their comment. We have now increased the size of the detections (represented as green circles) in the associated Suppl. Figs (now Supplemental Fig. 20 and 21, Supplementals pages 37-41).

Reviewer #4 (Remarks on code availability):

Two GitHub repositories are listed in the manuscript and accessible. However, the first repository (<https://github.com/Pathak-Lab/PIEZO1-LocalizationTools>) lacks a minimal example with raw data to test the code, and no hardware requirements or installation instructions are provided. The second repository (https://github.com/abcucberkeley/piezo1_analysis_pipeline) includes a minimal example, but hardware requirements are also missing. It appears that a high-performance workstation is needed, as I was unable to run the code on a small workstation with Windows 10 and only 16 GB RAM using Matlab R2023b.

We thank the reviewer for identifying this omission. We have completed the necessary edits to both GitHub repositories as described in our response to Reviewer #4 comment 5.

Response to reviewer comments

Reviewer #4 (Remarks to the Author):

Bertaccini et al. have addressed most of my comments and concerns in the revised manuscript and I support its publication in Nature Communications. I have only three minor comments/suggestions:

(1) Regarding the degree of labeling using JF-549, JF-646, and JF-635 HaloTag ligands: In the revised manuscript, the authors describe labeling with overnight incubation at 500 pM. In contrast, the JF-646-BAPTA-3AM probe was incubated for only 15 minutes. The authors state that they performed extensive optimization to achieve full labeling with low background of all three HaloTags of the PIEZO1-trimer. Standard live-cell HaloTag labeling protocols typically use 50–100 nM for 15–30 minutes at 37°C to achieve a high degree of labeling. Interestingly, the authors did not follow this approach and instead opted for much longer incubation times. Extended incubations at concentrations above 1 nM generally increase nonspecific binding. The authors should briefly explain their labeling strategy in the methods section and clarify why they did not use standard protocols. Additionally, they should mention in the text that the fluorogenicity of JF-646 (21-fold, Grimm et al., Nat Methods 12, 244–250 (2015)) and JF-635 (113-fold, Grimm et al., Nat Methods 14(10): 987–994 (2017)) was an important factor in reducing fluorescent background during long incubation times or when labeling complex samples such as MNRs, where deep tissue penetration requires extended incubation.

We thank the reviewer for their suggestion. We have now added a statement to the “HaloTag ligand treatment protocol” subsection in the Methods: “Upon testing a series of labeling conditions, we found that labeling overnight with a low concentration of Janelia Fluor 646 and Janelia Fluor 635 yielded clean labeling with low non-specific binding, likely due to the low expression level of endogenous PIEZO1 and high fluorogenicity of the JF-based probes^{45,71}. Labeling conditions are provided in Supplementary Table 2.”

(2) Some supplementary movies exhibit strong compression artifacts. These should be improved.

We thank the reviewer for their feedback. We have checked our supplementary movies and replaced the files showing compression artifacts.

(3) The authors stated that they replaced the terms motile/immotile with mobile/immobile throughout the manuscript. However, both versions still appear in multiple instances. This should be corrected for consistency.

We thank the reviewer for spotting this, and have corrected this.

Reviewer #4 (Remarks on code availability): Both GitHub repositories "PIEZO1-LocalizationTools" (<https://github.com/Pathak-Lab/PIEZO1-LocalizationTools>) and "piezo1_analysis_pipeline" (https://github.com/abcucberkeley/piezo1_analysis_pipeline) are accessible, provide a minimal example, hardware requirements and installation instructions. I could run the minimal example of the piezo1_analysis_pipeline using Matlab. The pipeline

seems to reproduce the data shown in Figure 5. The second toolbox PIEZO1-LocalizationTools provides enough information in the installation instructions and comprehensive guidelines for data processing. However, I was unable to install the FLIKA software, likely due to Python-related compatibility issues with my specific operating system and Python configuration.

We thank the reviewer for trying our analysis pipelines. We have since uploaded our repositories to a permanent DOI with Zenodo, which has now been cited in our manuscript “Data availability” section. We have also listed the software and associated version numbers in our “Code availability” section.